# From Parameters to Feature Space: Task Arithmetic for Backdoor Mitigation in Model Merging

Zhenqian Zhu [1]   Yamin Hu [1]   Yiya Diao [1]   Weixiang Li [2]   Haodong Li [2]   Wenjian Luo [1]

## Abstract

Model merging (MM) has gained significant attention as a cost-effective approach to integrate multiple task-specific models into a unified model. However, recent work reveals that MM is highly susceptible to backdoor attacks. Existing defenses based on task arithmetic often fail to eliminate backdoors without substantially degrading clean-task performance, owing to their reliance on direct parameter-space editing. To address this gap, we propose Linear Feature Path Minimization (LFPM), a backdoor mitigation framework for model merging, which introduces an anti-backdoor task vector into the backdoored merged model. Unlike prior approaches, LFPM formulates the backdoor robustness of the merged model from a unified feature-space perspective under the Cross-Task Linearity (CTL) framework, which leverages the approximate linearity of features across tasks. This perspective guides the optimization of the anti-backdoor task to suppress backdoors while preserving clean-task performance. Furthermore, we introduce an effective optimization mechanism based on gradient accumulation and loss path-integral, ensuring robust backdoor suppression along the interpolation path. Extensive experiments demonstrate that LFPM consistently exhibits strong robustness against backdoor attacks in both full fine-tuning and Parameter-Efficient Fine-Tuning (PEFT) settings.

[1]Institute of Cyberspace Security, School of Computer Science and Technology, Harbin Institute of Technology, Shenzhen, 518055, China. [2]Shenzhen Key Laboratory of Media Security, Shenzhen University, Shenzhen, 518060, China.. Correspondence to: Wenjian Luo <luowenjian@hit.edu.cn>.

*Proceedings of the $43^{rd}$ International Conference on Machine Learning*, Seoul, South Korea. PMLR 306, 2026. Copyright 2026 by the author(s).

## 1. Introduction

Pre-trained models (Dosovitskiy, 2020; Radford et al., 2021; Devlin et al., 2019; Touvron et al., 2023) constitute a fundamental component of modern machine learning systems, typically serving as backbones for downstream tasks. However, in multi-task scenarios, maintaining an individual model for each task incurs a significant computational overhead and leads to both knowledge redundancy and inefficient storage. To address this, Model Merging (MM) (Yadav et al., 2024) has emerged as a cost-effective technique that integrates multiple fine-tuned task-specific models into a single unified model, consolidating knowledge from foundation models across diverse domains. This approach naturally extends to a broad range of fine-tuning paradigms, including Parameter-Efficient Fine-Tuning (PEFT) (Han et al., 2024) such as Low-Rank Adaptation (LoRA) (Hu et al., 2022; Zanella & Ben Ayed, 2024), which enables effective task adaptation without updating the entire model.

Despite these advantages, recent studies (Zhang et al., 2024; Yin et al., 2024; Wang et al., 2025; Yuan et al., 2025) reveal that model merging introduces a new attack surface, particularly for backdoor attacks. In this setting, an adversary can implant a backdoor into the merged model by contributing only a compromised task-specific component, without requiring access to the overall merging process. This threat is further exacerbated in open-source platforms (Jain, 2022), where models are frequently downloaded, fine-tuned, and re-uploaded as reusable components. When merged-model developers incorporate modules obtained from third-party sources, these modules may potentially contain backdoors. As a result, such reliance on third-party model resources can facilitate the propagation of compromised components, thereby posing significant risks to the integrity and reliability of the resulting merged models.

However, only a few backdoor defense methods have been specifically developed for the model merging setting (Pawlak et al., 2025; Hsu C Y, 2025). Recent approaches (Pawlak et al., 2025; Hsu C Y, 2025) typically conceptualize backdoor behaviors as task vectors within a task arithmetic framework (Ilharco et al., 2022), where malicious backdoors are injected via vector addition, and backdoor mitigation through vector subtraction. This perspective

offers a new lens for understanding and designing backdoor attacks and defenses in model merging. Nevertheless, direct parameter-space editing (Pawlak et al., 2025) makes it challenging to achieve a favorable trade-off between maintaining clean performance and removing backdoors, due to the difficulty of disentangling backdoor-related parameters from those essential for clean task functionality.

Recent studies on Mode Connectivity (MC) (Garipov et al., 2018; Zhou et al., 2023; 2024) have revealed an intriguing linear phenomenon in the pretraining-finetuning paradigm, referred to as Cross-Task Linearity (CTL) (Zhou et al., 2024). Models initialized from the same pretrained checkpoint and fine-tuned on different tasks exhibit a near-linear behavior: when interpolating the parameters of two fine-tuned models, the features at each layer of the weight-interpolated model closely approximate the linear interpolation of the features from the individual models. Motivated by CTL, we investigate whether parameter-space task arithmetic can be approximated as controlled feature-space operations. This insight enables us to optimize an anti-backdoor task for backdoor mitigation from a unified feature-level perspective, thereby overcoming the limitations of existing defenses that rely directly on parameter-space interventions.

Building on this insight, we propose **Linear Feature Path Minimization (LFPM)**, a novel defense proposed for the model merging setting. *The core idea is to construct an anti-task vector via an anti-backdoor task, and merge it with a backdoored merged model to obtain a new purified merged model*. Notably, unlike existing methods that directly edit models in parameter space, LFPM guides the optimization of the anti-backdoor task from a unified feature-space perspective, achieving a principled balance between backdoor robustness and clean-task performance.

LFPM consists of two stages. **(1) Adversarial Feature Extraction via Subspace Partitioning.** In this stage, LFPM extracts adversarial feature vectors by partitioning the feature space, effectively disentangling adversarial perturbation features from clean semantic features. Unlike prior methods (Xu et al., 2024; Wei et al., 2023b), this approach does not require input-space trigger synthesis or adversarial perturbation generation. Instead, adversarial features are encoded via learned visual prompts (Zhou et al., 2022; Jia et al., 2022), thereby preserving the semantic integrity of the original samples. **(2) Anti-Backdoor Task Vector Generation.** We first reformulate the problem as a Sharpness-Aware Minimization (SAM) (Foret et al., 2020)-like optimization problem under the CTL framework, aiming to minimize sharpness (curvature) along the parameter interpolation path from the original backdoored merged model to anti-backdoor model. Based on the CTL framework, this is equivalent to minimizing sharpness along the feature interpolation path. Furthermore, feature deviations induced by CTL

are translated into worst-case perturbations in the prompt space, enhancing the suppression of backdoor behaviors. Moreover, we introduce a gradient accumulation approximation mechanism to facilitate the optimization objective. The formulation naturally supports a loss path-integral procedure. Together, these mechanisms ensure robust backdoor mitigation throughout the entire interpolation path. Overall, our contributions are as follows:

- **Feature-Space Anti-Backdoor Framework under CTL.** We propose a novel unified feature-space perspective for backdoor mitigation in model merging under the CTL framework, supported by both theoretical analysis and empirical validation.

- **Linear Feature Path Minimization.** We design a two-stage defense that effectively mitigates backdoor effects while preserving clean-task performance. The method eliminates the need for generating input-space adversarial perturbations or performing explicit parameter-space interpolation during training, and naturally supports the loss path-integral mechanism.

- **Extensive Empirical Validation.** We empirically demonstrate that LFPM consistently achieves strong backdoor mitigation across both full fine-tuning and PEFT settings such as LoRA, with stable performance along the parameter interpolation path.

## 2. Related Work

**Model Merging.** Model merging (Yadav et al., 2024; Yang et al., 2024) constructs a unified model by combining multiple task-specific models with distinct capabilities, without requiring access to the original training data. This approach not only improves task-specific performance but also significantly reduces computational overhead and storage inefficiency through effective knowledge transfer and inheritance. Existing approaches primarily focus on mitigating the performance degradation caused by task conflicts. Representative approaches include Simple Averaging (Wortsman et al., 2022), Task Arithmetic (Ilharco et al., 2022), Ties-Merging (Yadav et al., 2023), Della-Merging (Deep et al., 2024) among others.

**Backdoor Attacks and Defenses.** Zhang *et al.,* (Zhang et al., 2024) are the first to investigate backdoor attacks in model merging, revealing a novel attack surface where an adversary injects malicious backdoor behavior by contributing a compromised model. The merged model may inherit the embedded backdoor, which can be persistently triggered by the inputs containing the backdoor trigger on an adversary-specified downstream task. Yin et al. (Yin et al., 2024) further extend this threat model to the Parameter-Efficient Fine-Tuning (PEFT) setting, considering adversaries with constrained computational resources. Subse-

quently, several defenses tailored to model merging have been proposed (Pawlak et al., 2025). Recent works (Pawlak et al., 2025; Hsu C Y, 2025) further conceptualize backdoor behaviors as task vectors within the task arithmetic framework (Ilharco et al., 2022) and mitigate them via vector subtraction. However, directly editing parameters in the parameter space poses a challenge: it is difficult to balance clean-task performance with effective backdoor removal due to the difficulty of disentangling backdoor-related parameters from those essential for clean functionality.

**Mode Connectivity.** Mode Connectivity (MC) (Draxler et al., 2018; Freeman & Bruna, 2016) is a fundamental phenomenon in deep learning, where different minima in the loss landscape can be connected by a path of nearly constant loss. In certain cases, particularly when models are briefly co-trained before independent fine-tuning, these paths are approximately linear, forming Linear Mode Connectivity (LMC) (Garipov et al., 2018). Extending this notion, Layerwise Linear Feature Connectivity (LLFC) (Zhou et al., 2023) demonstrates that the linear relationship can hold at the feature level across each layer. Cross-Task Linearity (CTL) (Zhou et al., 2024) further generalizes LLFC by extending the linear relationship to models that share a common pretrained checkpoint but are independently fine-tuned on different downstream tasks. Formally, for downstream tasks $\mathbb{D}_i$ and $\mathbb{D}_j$ and layer $\ell$, $\forall \alpha \in [0,1]$,

$$f^{(\ell)}(\alpha\theta_i + (1-\alpha)\theta_j) \approx \alpha f^{(\ell)}(\theta_i) + (1-\alpha)f^{(\ell)}(\theta_j), \quad (1)$$

which indicates that features of the weight-interpolated model closely match the linear combination of features from individual models.

## 3. Preliminary

In this section, we first introduce the threat model for backdoor attacks in model merging from a task arithmetic perspective. We then present the defense objective of the defender, and reformulate it as a practical surrogate optimization. Finally, we analyze the stability factors of Cross-Task Linearity (CTL), which serve as the theoretical foundation for our work. For convenience, all notations used in this paper are summarized in Appendix A.

### 3.1. Threat Model

Let $f : \mathcal{X} \times \Theta \to \mathcal{Y}$ be a neural network that maps an input $x \in \mathcal{X}$ to an output in $\mathcal{Y}$, parameterized by weights $\theta \in \Theta$, where $\mathcal{X} \subseteq \mathbb{R}^d$, $\Theta \subseteq \mathbb{R}^m$, and $\mathcal{Y} \subseteq \mathbb{R}^c$. Given a pre-trained model with parameters $\theta_0$, we consider $k$ downstream tasks $\{\tau_i\}_{i=1}^k$, where each task $\tau_i$ is obtained by fine-tuning on its dataset $\mathcal{D}_i$, yielding task-specific parameters $\theta_i$. In particular, for CLIP-like pre-trained models $\mathcal{M} = \{\mathcal{V}, \mathcal{T}\}$ (Radford et al., 2021), fine-tuning typically updates only the visual encoder $\mathcal{V}$, while the text encoder $\mathcal{T}$ remains fixed

and defines a linear prediction head $h$ parameterized by $W$. Under this paradigm, each fine-tuned model can thus be decomposed as $f = h \circ g$, where $g$ denotes the task-adapted feature extractor (from $\mathcal{V}$), and $h$ is a task-specific linear prediction head. This decomposition provides a convenient framework for analyzing and mitigating backdoors in the feature space.

Following prior work on task arithmetic (Ilharco et al., 2022), each task is represented by a task vector $\Delta\theta_i = \theta_i - \theta_0$. Model merging combines these task vectors via a weighted combination to obtain a unified model that performs well across multiple tasks:

$$\theta_m = \theta_0 + \Delta\theta_m = \theta_0 + \sum_{i=1}^{k} \lambda_i \Delta\theta_i \quad (2)$$

where $\Delta\theta_m = \sum_{i=1}^{k} \lambda_i \Delta\theta_i$ denotes the merged task vector and $\{\lambda_j\}$ are task-specific merging coefficients.

**Adversary.** We consider a threat model where an adversary controls a single task $\tau_j$, referred to as the **adversary task**, by injecting backdoor samples into its fine-tuning dataset $\mathcal{D}_j$. As a result, the adversary task vector can be decomposed as $\Delta\theta_j = \Delta\theta_{c_j} + \Delta\theta_b$, where $\Delta\theta_{c_j}$ captures clean task-specific adaptation, $\Delta\theta_b$ encodes backdoor-related behaviors. Importantly, the adversary does not have direct access to other tasks or the overall merging process. The adversary aims to attack a specific downstream task, termed the **target task**. Through model merging, the backdoor component propagates into the merged task vector, resulting in $\Delta\theta_m = \Delta\theta_c + \Delta\theta_b$, where $\Delta\theta_c$ denotes the clean aggregated component from all tasks. Accordingly, the resulting backdoored merged model is given by

$$\theta_m = \theta_0 + \Delta\theta_c + \Delta\theta_b. \quad (3)$$

The adversary's objective is to ensure that, at inference time, triggered inputs are misclassified into an adversary-chosen target class for the target task.

### 3.2. Problem Formulation

**Defender.** From the defender's perspective, typically the merged-model developer, a collection of task vectors $\{\Delta\theta_i\}_{i=1}^k$ is available, among which some may be contaminated by backdoor components. The defender has no prior knowledge of which task vectors are malicious. The objective is to mitigate backdoor effects in the merged model $\theta_m$ while preserving clean-task performance. We assume the defender has access to a shadow dataset $\mathcal{D}_s$. An *anti-backdoor task* is constructed by generating adversarial examples $x^{'} = x + \Delta x$ from $\mathcal{D}_s$, yielding the dataset $\mathcal{D}_{adv}$, which is then treated as the $(k+1)$-th task $\tau_{k+1}$ in the original task sequence. The anti-backdoor task vector $\Delta\theta_{k+1}$ is obtained by fine-tuning the pre-trained model on the combined dataset $\mathcal{D}_s \cup \mathcal{D}_{adv}$. As illustrated in Fig. 1(a), this task

vector is incorporated into the merged model via additive task arithmetic, yielding a new merged model:

$$\boldsymbol{\theta}_m' = \boldsymbol{\theta}_0 + (1-\alpha)\Delta\boldsymbol{\theta}_m + \alpha\Delta\boldsymbol{\theta}_{k+1} \quad (4)$$

where $\alpha \in [0.0, 1.0]$ controls the strength of intervention.

**Defense Objective.** The defender aims to: (i) effectively eliminate the backdoor effect, (ii) preserve clean-task performance, and (iii) achieve consistent robustness along the entire parameter interpolation path. Formally, let $\mathcal{D}_c = \bigcup_{i:k} \mathcal{D}_i$ denotes the union of clean datasets, and $\mathcal{D}_b$ denotes the unknown backdoor dataset. Then, for any $\alpha \in [0.0, 1.0]$, the defense objective is formulated as:

$$\begin{cases} \min_{\Delta\boldsymbol{\theta}_{k+1}} \mathbb{E}_{(x,y)\sim\mathcal{D}_c}[\mathcal{L}(f(x;\boldsymbol{\theta}_m'),y)], \\ \max_{\Delta\boldsymbol{\theta}_{k+1}} \mathbb{E}_{(x_b,y_t)\sim\mathcal{D}_b}[\mathcal{L}(f(x_b;\boldsymbol{\theta}_m'),y_t)], \end{cases} \quad (5)$$

where $(x_b, y_t)$ denote backdoor inputs and target labels. This formulation captures the goal of minimizing clean-task risk while maximizing backdoor risk.

Directly optimizing Eq. 5 is infeasible, as the defender has no access to the backdoor dataset $\mathcal{D}_b$. Existing post-training defenses for single-model (Xu et al., 2024; Wei et al., 2023b) typically address this challenge by approximating backdoor inputs via trigger inversion or by replacing the backdoor risk with an adversarial risk.

**Optimization Objective.** Following prior work (Wei et al., 2023b), we replace the backdoor risk with the adversarial risk and reformulate the defense objective as a surrogate optimization problem. For any $\alpha \in [0.0, 1.0]$, we have:

$$\begin{cases} \min_{\Delta\boldsymbol{\theta}_{k+1}} \mathbb{E}_{(x,y)\sim\mathcal{D}_s}[\mathcal{L}(f(x;\boldsymbol{\theta}_m'),y)], \\ \min_{\Delta\boldsymbol{\theta}_{k+1}} \mathbb{E}_{(x',y)\sim\mathcal{D}_{adv}}[\mathcal{L}(f(x';\boldsymbol{\theta}_m'),y)]. \end{cases} \quad (6)$$

However, such direct replacement can degrade clean-task performance due to the substantial gap between adversarial and backdoor samples, as reported in prior studies (Wei et al., 2023b; Gao et al., 2023). To mitigate this, we require the surrogate samples $x'$ to exhibit feature-level characteristics similar to those of backdoor samples: (i) activating backdoor-related features in the backdoored merged model, and (ii) preserving clean semantic features in the purified model. Formally, these conditions are expressed as

$$\begin{cases} f(x';\boldsymbol{\theta}_m) \approx f(x_b;\boldsymbol{\theta}_m) \\ f(x';\boldsymbol{\theta}_m') \approx f(x;\boldsymbol{\theta}_m'). \end{cases} \quad (7)$$

These conditions ensure that the surrogate optimization preserves clean-task performance.

### 3.3. Deviation Bound for Cross-Task Linearity

To bridge the optimization objective in Eq. 6 from the parameter-interpolated path to the feature-interpolated path, we derive the CTL deviation bound and analyze its stability factors, which form the theoretical foundation of our

method. The proof of Theorem 3.1 is in Appendix C.1.

**Theorem 3.1** (**Deviation Bound for CTL**). *Let $f : \mathbb{R}^d \rightarrow \mathbb{R}^c$ be a twice continuously differentiable function defined on an open convex set $\Theta \subseteq \mathbb{R}^m$. Consider two fine-tuned models $\boldsymbol{\theta}_i, \boldsymbol{\theta}_j \in \Theta$ originating from the same pretrained checkpoint, and define the linear interpolation path*

$$\boldsymbol{\theta}(t) = (1-t)\boldsymbol{\theta}_i + t\boldsymbol{\theta}_j = \boldsymbol{\theta}_i + t\delta, \quad (8)$$

*where $\delta := \boldsymbol{\theta}_j - \boldsymbol{\theta}_i$ and $t \in [0,1]$.*

*For any $\alpha \in (0,1)$, define the feature deviation for CTL as $\Delta(\alpha) := f(\boldsymbol{\theta}(\alpha)) - (1-\alpha)f(\boldsymbol{\theta}_i) - \alpha f(\boldsymbol{\theta}_j)$. Then, the norm of $\Delta(\alpha)$ admits the path-integral form*

$$\|\Delta(\alpha)\| = \left\| \int_0^1 k_\alpha(t)\delta^T \bigtriangledown^2 f(\boldsymbol{\theta}(t))\delta dt \right\|. \quad (9)$$

*where $k_\alpha(t) = \begin{cases} (1-\alpha)t, 0 \le t \le \alpha, \\ \alpha(1-t), \alpha \le t \le 1. \end{cases}$*

*Moreover, the deviation is bounded by*

$$\|\Delta(\alpha)\| \le \|\delta\|^2 \left[ (1-\alpha)\int_0^\alpha tH(t)dt + \alpha\int_\alpha^1 (1-t)H(t)dt \right] \quad (10)$$

*where $H(\theta(t)) = \bigtriangledown^2 f(\theta(t))$ denotes the Hessian along the interpolation path.*

*Remark.* For simplicity, we treat the vector-valued output of $f$ as scalar-valued; all results hold element-wise for vector-valued outputs, including Corollary 4.2.

Theorem 3.1 provides a path-integral formulation of the feature deviation for CTL. When the Hessian along the interpolation path is uniformly bounded, *i.e.,* $\|H(\theta(t))\| \le \lambda_{max}$, the deviation admits the simplified bound

$$\|\Delta(\alpha)\| \le \frac{\alpha(1-\alpha)\lambda_{max}\|\delta\|^2}{2} \quad (11)$$

which aligns with prior results (Zhou et al., 2024).

This bound highlights key factors governing CTL stability:

- **The parameter distance** $\|\delta\|$ between two finetuned models, reflecting the magnitude of parameter-space shift induced by task-specific adaptation.

- **The landscape flatness**, captured by Hessian bound $H(t)$ along the interpolation path $\{\theta(t)\}_{t\in[0,1]}$.

- **The merging coefficient** $\alpha$, which modulates the contribution of curvature via the weight function $k_\alpha(t)$, peaking at $\alpha = 0.5$ (*i.e.,* $\int_0^1 k_{0.5}(t) = 0.25$).

In practice, task-specific fine-tuning inevitably increases $\|\delta\|$, which may violate CTL under large parameter shifts. To mitigate this, Theorem 3.1 suggests two complementary strategies: **Biased merging**, which selects a $\alpha$ closer to an endpoint to reduce $k_\alpha(t)$ and tighten the deviation bound; **Landscape smoothing**, which reduces the effective curvature $H(\theta(t))$ along the interpolation path to preserve

near-linear behavior over a larger portion of the path.

## 4. Methodology

As illustrated in Fig. 1(a), we aim to achieve consistent robustness along the parameter interpolation path between $\theta_m$ and $\theta_{k+1}$, as defined by the optimization objective in Eq. 6. Within the CTL framework, this objective can be equivalently reformulated in feature space, *i.e.,* ensuring robustness along the feature interpolation path from $z_m$ to $z_{k+1}$, as shown in Fig. 1(b). This provides a unified feature-space perspective for backdoor mitigation in the merged model, as supported by empirical validation in Appendix E.5.

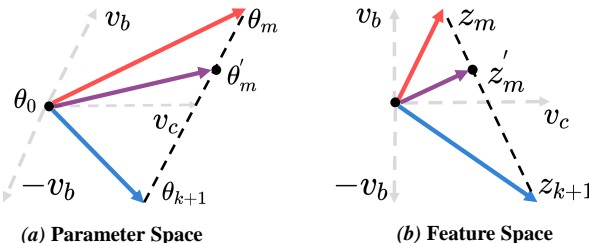

*(a) Parameter Space*      *(b) Feature Space*

*Figure 1.* The core idea of LFPM. $v_c$ and $v_b$ denote the directions of the clean-task and backdoor-task vectors, respectively, with $-v_b$ indicating the reversed backdoor direction. $\theta_m$ and $\theta_{k+1}$ denote the merged model and the anti-backdoor model, while $z_m$ and $z_{k+1}$ are their respective feature representations. Under the CTL framework, robustness along the parameter interpolation path from $\theta_m$ to $\theta_{k+1}$ can be achieved by performing robust optimization along the feature interpolation path from $z_m$ to $z_{k+1}$.

Building on this insight, we propose **Linear Feature Path Minimization (LFPM)**, a defense framework for model merging, consisting of two stages: **(i) Adversarial Feature Extraction via Subspace Partitioning:** this stage extracts adversarial features by disentangling adversarial representations from clean semantic features through feature subspace partitioning; **(ii) Anti-backdoor Task Vector Optimization:** this stage optimizes an anti-backdoor task vector to ensure robust backdoor mitigation along the interpolation path. The overall LFPM algorithm is presented in Appendix B.1, with detailed procedures for each stage provided in Appendices B.2 and B.3, respectively.

In addition, we conduct a curvature analysis throughout training dynamics using Hessian-Vector Products (HVPs) (Dagreou et al., 2024) to understand the underlying principles of LFPM, with results reported in Appendix E.6.

### 4.1. Adversarial Feature Extraction via Subspace Partitioning

Based on the constraint conditions of the surrogate optimization objective in Eq. 7, LFPM aims to extract adversarial features while preserving clean-task semantics. To this end, we first establish the following principle, which character-

izes the manifestation of adversarial perturbations in the feature space, with the proof provided in Appendix C.2.

**Lemma 4.1** (**Adversarial Perturbation Orthogonality Principle**). *Let $f = h \circ g$, where $g$ is a feature extractor and $h(z) = W^\top z + b$ is a linear prediction head for a downstream task. Suppose an adversarial input $\boldsymbol{x}' = \boldsymbol{x} + \Delta\boldsymbol{x}$ preserves the clean-task semantics,* i.e.,

$$f(x'; \boldsymbol{\theta}'_m) = f(x; \boldsymbol{\theta}'_m). \tag{12}$$

*Then, under a first-order approximation of $g$, the induced feature perturbation $\Delta z = g(x') - g(x)$ must satisfy*

$$\Delta z \perp \mathrm{span}(W), \tag{13}$$

*where $\mathrm{span}(W)$ denotes the feature subspace spanned by the clean-task feature vectors.*

This lemma highlights a key property: to preserve clean semantic features, adversarial perturbations must lie in directions orthogonal to the clean semantic subspace. Based on Lemma 4.1, we extract adversarial features via subspace partitioning, which consists of two steps: feature subspace partitioning and prompt-based adversarial feature mining.

**Feature Subspace Partitioning**. We define a rank-$r$ projection matrix $P_s \in \mathbb{R}^{n \times n}$ associated with adversarial features, projecting onto the adversarial subspace $S$,

$$P_s = UU^T \tag{14}$$

where $U \in \mathbb{R}^{n \times r}$ is a semi-orthogonal matrix satisfying $U^\top U = I_{r \times r}$. By learning $P_s$, the feature space $V$ can be decomposed into two orthogonal subspaces: an adversarial feature subspace $S$ and a clean feature subspace $T$, such that $V = S \oplus T$. Accordingly, the projection matrix for the clean feature subspace $T$ is $P_t = I - P_s$.

For a given input $x$, let $z \in V$ denote its feature representation. Based on the projection matrix $P_s$, $z$ can be decomposed into an adversarial component $P_s z$ and a clean component $P_t z$, where $P_s z$ captures task-irrelevant or adversarial directions, while $P_t z$ preserves the discriminative semantic features corresponding to the ground-truth label $y$. Accordingly, for the merged model $f(\theta_m) = h_m \circ g_m$, feature subspace partitioning is formulated as:

$$\min_U \sum_{(x,y)\in\mathcal{D}_s} \left[ \mathcal{L}\big(h_m(P_t z), y\big) - \mathcal{L}\big(h_m(P_s z), y\big) + \mathcal{R}(U) \right] \tag{15}$$

where $z = g_m(x)$, and $\mathcal{R}(U) = \left\| U^T U - I_{r \times r} \right\|$ enforces the orthogonality of $U$.

**Prompt-Based Adversarial Feature Mining.** Unlike existing approaches (Wei et al., 2023b; Gao et al., 2023) that directly generate adversarial perturbations or reverse-engineer triggers in the input space, our method encodes adversarial perturbations using *learnable visual prompts $P$* (Zhou et al., 2022; Jia et al., 2022), thereby preserving the semantic integrity of the original input $x$.

To satisfy the surrogate condition in Eq. 7, we leverage two characteristic properties of backdoor models (Wei et al., 2023a): **Backdoor-induced output deviation:** Backdoor triggers induce significant output deviations for the backdoored model, *i.e.*, $f(x; \theta_m) \neq f(x_b; \theta_m)$. To capture this effect, we amplify the discrepancy between the adversarial component of the perturbed input $x^{'} = P \oplus x$ and that of the original input $x$; **Compactness of backdoor-related features:** Backdoor features tend to cluster tightly in feature space, which motivates enforcing batch-wise consistency among adversarial components. Simultaneously, to preserve clean-task semantics, the clean component of $x^{'}$ is aligned with that of the original input $x$.

Accordingly, these constraints are naturally incorporated into the adversarial feature mining objective:

$$\min_{P} \sum_{(x,y) \in D_s} \text{sim}(P_t z_c^{'}, P_t z_c) - \text{sim}(P_s z_m^{'}, P_s z_m) + \left\| P_s z_m^{'} - \mu_s \right\| \quad (16)$$

where $z_c, z_c^{'}$ are features of a clean reference model, approximated in practice using a pre-trained model, and $z_m, z_m^{'}$ are from the backdoored merged model, with $z'$ corresponding to input $x^{'}$. $\mu_s = \mathbb{E}_{x \sim \mathcal{B}}(P_s z_m^{'})$ is the batch-wise mean adversarial feature. $\text{sim}(\cdot, \cdot)$ measures the similarity between projected features (*e.g.,* mean squared error).

The three terms in Eq. 16 enforce clean semantic preservation, clean-adversarial feature separation in the subspace $S$, and compact clustering of adversarial components, respectively. Collectively, they ensure that $x^{'}$ activates adversarial features while preserving the main semantic features, balancing backdoor mitigation with clean-task performance.

### 4.2. Anti-backdoor Task Vector Optimization

In this subsection, we first reformulate the original robustness optimization of the interpolated model into a SAM-like optimization form within the CTL framework. We then recast it as an equivalent optimization along the feature interpolation path, and map CTL-induced feature deviations to worst-case perturbations in the prompt space. Building on this formulation, we introduce gradient accumulation optimization to efficiently solve the resulting objective, along with the concept of a loss path-integral, enabling robust backdoor suppression throughout the interpolation path.

**SAM-like Optimization.** Leveraging the CTL feature deviation bound from Theorem 3.1, the adversarial optimization in Eq. 6 can be reformulated under the CTL constraint as:

$$\min_{\Delta\theta_{k+1}} \mathcal{L}\big(f(x^{'}, \theta_m + \delta(\alpha)); y\big) \quad \text{s.t.} \quad \|\Delta(\alpha)\| \leq \varepsilon, \quad (17)$$

where $\theta_m^{'} = \theta_m + \delta(\alpha)$, $\delta(\alpha) = \alpha(\Delta\theta_{k+1} - \Delta\theta_m)$ is the scaled update along the interpolated path, and $\|\Delta(\alpha)\|$ denotes feature deviation induced by the CTL.

Since the merged model $\theta_m$ is fixed, $\mathcal{L}(f(x^{'}; \theta_m); y)$ is

constant. To explicitly highlight the sharpness term, we can rewrite the constrained optimization as

$$\min_{\Delta\theta_{k+1}} \max_{|\Delta(\alpha)| \leq \varepsilon} \mathcal{L}(f(x^{'}; \theta_m + \delta(\alpha)); y) - \mathcal{L}(f(x^{'}; \theta_m); y). \quad (18)$$

The term captures the sharpness of $\mathcal{L}$ along the parameter interpolation path. This objective seeks a parameter point $\theta_{k+1}$ (with $\|\Delta(\alpha)\| \leq \varepsilon$) such that the loss decreases most rapidly along the path from $\theta_m$ to $\theta_{k+1}$. This resembles the Sharpness-Aware Minimization (SAM) framework (Foret et al., 2020), where sharpness is reduced via worst-case weight perturbations in the neighborhood of the current parameters. The difference lies in the fact that, here, the perturbations are restricted to the interpolation path.

**From Parameter Interpolation to Feature Interpolation.** Under the CTL framework, for any $\alpha \in [0.0, 1.0]$, we have: $f(\theta(\alpha)) \approx (1 - \alpha) f(\theta_i) + \alpha f(\theta_j)$. By re-expressing the feature deviation as $\Delta z = \Delta(\alpha)$, the original SAM-like optimization along the parameter interpolation path can be equivalently reformulated as robust optimization along the feature interpolation path:

$$\min_{\Delta\theta_{k+1}} \max_{|\Delta z| \leq \rho} \mathcal{L}\big(h\big((1 - \alpha)g(x^{'}; \theta_m) + \alpha g(x^{'}; \theta_{k+1}) + \Delta z\big); y\big) \quad (19)$$

where $(1 - \alpha)g(x^{'}; \theta_m) + \alpha g(x^{'}; \theta_{k+1})$ denotes the interpolation of the feature outputs from the feature extractor $g$, and $\Delta z \in \{v : ||v|| \leq \rho\}$, which represents the allowable range of feature-space perturbations.

**Mapping Feature Deviation to Prompt Perturbation.** As described in Section 4.1, adversarial perturbations are encoded via learnable visual prompt $P$, where the perturbed input $x^{'} = P \oplus x$ corresponds to a feature representation $z = g(P \oplus x; \theta)$. Building on this formulation, we further map feature deviations induced by CTL to worst-case perturbations in the prompt space, resulting in the following optimization problem:

$$\min_{\Delta\theta_{k+1}} \max_{|\Delta P| \leq r_\rho} \mathcal{L}[h\big((1 - \alpha)g(x^{''}; \theta_m) + \alpha g(x^{''}; \theta_{k+1})\big); y] \quad (20)$$

where $x^{''} = (P + \Delta P) \oplus x$ and the prompt perturbation $\Delta P$ is bounded as $r_\rho \leq \frac{\rho}{\|J_P\|_2}$, and $J_P = \frac{\partial z}{\partial P}\big|_{P=P_0}$ denotes the Jacobian matrix of feature representation $z$ with respect to the learnable prompt $P$. A formal proof is provided in the Appendix C.3.

In summary, the above combination transforms the problem into a tractable form, eliminating the need to generate input-space adversarial perturbations or explicit parameter-space interpolation during iterative optimization, while shifting the search for worst-case perturbations from the high-dimensional parameter space to the low-dimensional prompt space. We now introduce gradient accumulation optimization to solve the resulting objective.

**Gradient Accumulation Approximation.** To approximate the gradient of an interpolated model $\theta_m^{'}$ using only the

endpoint $\boldsymbol{\theta}_m$ and $\boldsymbol{\theta}_{k+1}$, we state the following corollary.

**Corollary 4.2 (Gradient Approximation Along Linear Interpolation).** *Under the setting of Theorem 3.1, assume that along the interpolation path $\boldsymbol{\theta}(t)$,*

$$\left\| \nabla^2 f(\boldsymbol{\theta}(t)) \right\| \leq \lambda_{\max}, \quad \left\| \nabla_{\boldsymbol{\theta}} \nabla^2 f(\boldsymbol{\theta}(t)) \right\| \leq J_{\max}. \quad (21)$$

*Let $\boldsymbol{\theta}(\alpha) = (1-\alpha)\boldsymbol{\theta}_i + \alpha\boldsymbol{\theta}_j$ for $\alpha \in (0,1)$ and $\delta := \boldsymbol{\theta}_j - \boldsymbol{\theta}_i$.*

*Then, for any $\alpha \in (0,1)$, the gradient at the interpolated point $\boldsymbol{\theta}(\alpha)$ can be approximated by a convex combination of the endpoint gradients,*

$$\nabla f(\boldsymbol{\theta}(\alpha)) = (1-\alpha)\nabla f(\boldsymbol{\theta}_i) + \alpha\nabla f(\boldsymbol{\theta}_j) + \xi(\alpha), \quad (22)$$

*where the approximation error $\xi(\alpha)$ satisfies*

$$\|\xi(\alpha)\| \leq 2\alpha(1-\alpha)\lambda_{\max} \|\delta\| + \frac{\alpha(1-\alpha)}{2} J_{\max} \|\delta\|^2. \quad (23)$$

The proof is provided in Appendix C.4.

Corollary 4.2 provides a theoretical guarantee that gradients along the interpolation path can be approximated by the endpoint gradients with a bounded error. In practice, rather than computing the exact interpolated gradient, we update $\theta_{k+1}$ using a gradient accumulation strategy:

$$\boldsymbol{\theta}_{k+1} \longleftarrow \boldsymbol{\theta}_{k+1} - \eta\big(\nabla_{\boldsymbol{\theta}_{k+1}}\mathcal{L} + \lambda\nabla_{\boldsymbol{\theta}_m}\mathcal{L}\big) \quad (24)$$

where $\eta$ is the learning rate, and $\lambda$ is the scaling factor controlling the contribution of gradient from $\boldsymbol{\theta}_m$ in the accumulated update of $\boldsymbol{\theta}_{k+1}$.

**Loss Path-Integral.** The feature interpolation optimization form in Eq. 19 naturally admits the path-integral loss:

$$\int_{\alpha_2}^{1} \mathcal{L}\big(f(x, \boldsymbol{\theta}'_m), y\big)dt. \quad (25)$$

where $\alpha_2$ defines the interpolation intervals of interest along the path. This formulation is independent of the specific input $x$ (whether perturbed or unperturbed).

By leveraging feature-space interpolation, this integral can be approximated with a discrete sum:

$$\frac{1}{N}\sum_{i=i_2}^{N} \mathcal{L}\big(h\big((1-t_i)g(x'; \theta_m) + t_i g(x'; \theta_{k+1})\big); y\big) \quad (26)$$

where the the interval $[0.0, 1.0]$ is uniformly divided into $N$ sub-intervals of step size $\Delta t = 1/N$, giving discrete points $t_i = i\Delta t$ for $i = 0, 1, \ldots, N$. The index corresponding to the bound $\alpha_2$ is set as $i_2 = \lfloor \alpha_2 N \rfloor$.

## 5. Experiments

**Datasets and Models.** Following (Zhang et al., 2024), we adopt CLIP-ViT-B/32 as the default backbone and fine-tune task-specific models on 11 datasets (Kornblith et al., 2019): CIFAR100, Cars196, SUN397, EuroSAT, GTSRB, Pets, CIFAR10, SVHN, STL-10, Food-101, RESISC45. In addi-

tion, we randomly sample 10,000 images from ImageNet-1K (Deng et al., 2009) to form the shadow dataset for anti-backdoor task optimization. Six distinct tasks are randomly selected to form a task sequence, with the first task as the adversary task and the second as the target task. All task sequences and experimental settings are summarized in the Appendix D.1. Task Sequence 1 is used by default, while results for the other sequences are reported in Appendix E.3.

**Baselines. (i) Attack Baselines:** To account for adversaries with varying resource budgets, we consider Bad-Merging (Zhang et al., 2024) under full fine-tuning and LoBAM (Yin et al., 2024) under parameter-efficient fine-tuning (PEFT) using LoRA (Hu et al., 2022; Zanella & Ben Ayed, 2024). **(ii) Defense Baselines:** We evaluate representative backdoor defense methods from complementary perspectives. Specifically, these methods include IBVS (Pawlak et al., 2025), which mitigates backdoors via task arithmetic, and SAU (Wei et al., 2023b), a state-of-the-art post-training defenses based on adversarial training. We also consider sharpness-aware defenses, namely SAM (Foret et al., 2020) and PAM (Min et al., 2024), which improve robustness by minimizing loss sharpness through weight perturbations. **(iii) Model Merging (MM) algorithm:** We evaluate standard MM methods, including Task-Arithmetic (Ilharco et al., 2022), Simple Average (Wortsman et al., 2022),Ties-Merging (Yadav et al., 2023), Della-Merging (Deep et al., 2024). Task-Arithmetic (TA) is used as the default merging strategy, with results for alternative algorithms reported in Appendix E.1. Details for attacks and defenses are provided in Appendices D.2 and D.3

**Evaluation Metric.** We evaluate merged models using clean accuracy (CA) and attack success rate (ASR), aiming for higher CA and lower ASR for the purified merged model. Specifically, we consider three complementary perspectives. **Target Task**: The effectiveness of attacks on the target task is measured by ASR(T). **Non-Target Task**: cross-task backdoor effect is quantified by ASR(N) = $\mathbb{P}(f(x) \neq f(x_b))$, capturing how a trigger designed for the target task affects predictions on non-target tasks. **Persistence:** We analyze ASR(T) along the interpolation path between the backdoored merged model and the purified model. Specifically, *Path-ASR* is defined as: Path-ASR = $\int_0^1 \text{ASR}(\alpha)d\alpha$, which measures the cumulative backdoor vulnerability along the interpolation path. A lower Path-ASR indicates more consistent mitigation performance, even when the model is closer to the backdoored source (i.e., smaller $\alpha$). In addition, unlike prior works that evaluate robustness at a single interpolation coefficient $\alpha$, Path-ASR provides a more comprehensive assessment by accounting for the practical uncertainty of model merging, where the optimal coefficient may vary across tasks and deployment settings. [1]

---

[1]For clarity, adversary tasks are marked in **blue** and target tasks

*Table 1.* Backdoor Defense Performance under BadMerging (ACC↑ / ASR ↓).

| Task → | CIFAR100 | | Cars196 | | SUN397 | | EuroSAT | | GTSRB | | Pets | | Average | | |
|---|---|---|---|---|---|---|---|---|---|---|---|---|---|---|---|
| Method ↓ | CA | ASR(N) | CA | ASR(T) | CA | ASR(N) | CA | ASR(N) | CA | ASR(N) | CA | ASR(N) | CA | ASR(T) | ASR(N) |
| Individual | 83.56 | 75.18 | 64.05 | 79.31 | 71.53 | 56.37 | 97.83 | 30.61 | 94.27 | 49.46 | 88.11 | 42.60 | 83.22 | 79.31 | 55.58 |
| TA | 68.03 | 69.12 | 58.11 | 98.42 | 65.03 | 78.26 | 62.16 | 50.40 | 43.05 | 91.57 | 85.39 | 58.87 | 63.62 | 98.42 | 69.64 |
| IBVS | 36.87 | 61.69 | 44.75 | 64.18 | 56.44 | 63.37 | 22.53 | 64.05 | 16.66 | 68.90 | 79.85 | 34.66 | 42.85 | 64.18 | 58.53 |
| SAU | 53.92 | 75.59 | 43.98 | 63.00 | 60.55 | 63.14 | 15.40 | 77.16 | 27.64 | 78.15 | 80.51 | 25.64 | 47.00 | 63.00 | 63.93 |
| SAM | 40.61 | 31.14 | 34.15 | 10.03 | 55.76 | 21.76 | 20.87 | 47.50 | 16.95 | 47.33 | 74.97 | 8.36 | 40.55 | 10.03 | 31.21 |
| PAM | 51.91 | 20.20 | 43.42 | 4.42 | 58.66 | 17.10 | 21.57 | 27.77 | 28.50 | 38.17 | 76.31 | 6.73 | 46.72 | 4.42 | 21.99 |
| LFPM | **68.00** | **10.38** | **55.15** | **0.49** | **64.88** | **7.65** | **45.66** | **17.51** | **33.28** | **20.32** | **85.06** | **2.75** | **58.67** | **0.49** | **9.85** |

*Table 2.* Backdoor Defense Performance under LoBAM (ACC↑ / ASR ↓).

| Task → | CIFAR100 | | Cars196 | | SUN397 | | EuroSAT | | GTSRB | | Pets | | Average | | |
|---|---|---|---|---|---|---|---|---|---|---|---|---|---|---|---|
| Method ↓ | CA | ASR(N) | CA | ASR(T) | CA | ASR(N) | CA | ASR(N) | CA | ASR(N) | CA | ASR(N) | CA | ASR(T) | ASR(N) |
| Individual | 80.17 | 84.08 | 61.65 | 54.33 | 67.93 | 48.38 | 96.87 | 27.90 | 89.76 | 21.25 | 89.56 | 18.28 | 80.99 | 54.33 | 39.97 |
| TA | 64.48 | 53.65 | 56.78 | 86.12 | 63.32 | 71.15 | 44.48 | 39.72 | 33.88 | 90.44 | 87.27 | 40.09 | 58.36 | 86.12 | 59.01 |
| IBVS | 37.28 | 31.99 | 40.35 | 8.56 | 47.20 | 59.42 | 24.40 | 59.25 | 11.67 | 78.70 | 71.76 | 34.36 | 38.77 | 8.56 | 52.74 |
| SAU | 50.96 | 83.91 | 44.43 | 7.30 | 57.10 | 71.53 | **39.40** | 61.03 | 22.18 | 92.46 | 81.35 | 24.69 | 49.23 | 7.30 | 66.72 |
| SAM | **60.57** | 19.06 | 53.26 | 3.18 | 61.83 | 21.08 | 34.98 | 38.07 | **25.49** | 39.47 | 85.88 | 8.17 | 53.66 | 3.18 | 25.17 |
| PAM | 57.80 | 21.20 | 51.18 | **3.07** | 60.68 | 21.33 | 33.83 | **31.98** | 25.48 | **39.23** | 84.21 | **7.71** | 52.19 | **3.07** | **24.29** |
| LFPM | 62.22 | **18.79** | **53.53** | 3.84 | **61.91** | **18.36** | 38.31 | 37.61 | 24.80 | 50.26 | **85.96** | 8.39 | **54.45** | 3.84 | 26.68 |

**Additional Experiments.** We provide a comprehensive set of additional experiments in the appendix, including generality across model merging algorithms (Appendix E.1), robustness under multiple compromised models (Appendix E.2), robustness under different task combinations (Appendix E.3), scalability to larger architectures (Appendix E.4), empirical validation of CTL (Appendix E.5), curvature analysis (Appendix E.6), computational cost comparison (Appendix E.7), efficiency analysis (Appendix E.8), and ablation studies (Appendix E.9).

### 5.1. Defense Performance

From Tables 1 and 2, we observe that, compared with individual models, backdoor triggers substantially alter the predictions of merged models on non-target tasks. Specifically, the average ASR(N) increases from 55.58% to 69.64% under BadMerging and from 39.97% to 59.01% under LoBAM. This effect is further amplified for alternative task sequences, as reported in Appendix E.3. For example, under BadMerging for Task sequence 3, the average ASR(N) increases from 32.29% to 88.09%. These results indicate pronounced cross-task backdoor leakage, where target-task triggers propagate to non-target tasks through model merging.

IBVS struggles to balance backdoor mitigation and model performance, with a high average ASR(T) of 64.18% and an average ASR(N) of 58.53% under BadMerging, while

in red in each task sequence. Bold and underlined values indicate the best and second-best results, respectively.

incurring substantial drops in average CA ($\sim$ 20.77%). This can be attributed to the challenge of disentangling clean-task parameters from backdoor-related ones. SAU performs well against LoBAM, but fails when full fine-tuning in BadMerging, resulting in the average ASR(T) of 63.00% and the average ASR(N) of 63.93%.

Under the LoBAM, SAM and PAM achieve relatively low average ASR(T) and ASR(N), while LFPM achieves comparable average ASR and slightly higher average CA. However, LFPM consistently outperforms them under BadMerging, achieving the lowest ASR(T) and ASR(N), as well as the highest CA across all tasks. In particular, LFPM outperforms SAM and PAM in average CA by 18.12% and 11.95%, respectively. Overall, these results demonstrate that LFPM effectively mitigates backdoor effects while preserving clean-task performance in both full fine-tuning and PEFT settings.

### 5.2. Robustness Evaluation along the Parameter Path

Rather than evaluating the robustness of a merged model at a single point, the *Path-ASR* provides a quantitative measure of the cumulative backdoor effect along the interpolation path. The results are presented in Fig. 2.

As shown in Fig. 2, SAM-like methods, including SAM, PAM, and our proposed LFPM, exhibit strong overall robustness, reflected by a rapid decline in ASR(T) along the interpolation path. Notably, LFPM exhibits the smallest Path-ASR (*i.e.,* the area under the ASR curve) under both

*Table 3.* Backdoor Defense Performance under BadMerging with Adaptive Attack (ACC↑ / ASR ↓).

| Task → | CIFAR100 | | Cars196 | | SUN397 | | EuroSAT | | GTSRB | | Pets | | Average | | |
|---|---|---|---|---|---|---|---|---|---|---|---|---|---|---|---|
| Method ↓ | CA | ASR(N) | CA | ASR(T) | CA | ASR(N) | CA | ASR(N) | CA | ASR(N) | CA | ASR(N) | CA | ASR(T) | ASR(N) |
| Individual | 84.21 | 98.85 | 64.05 | 79.31 | 71.53 | 56.37 | 97.83 | 30.61 | 94.27 | 49.46 | 88.11 | 42.60 | 83.22 | 79.31 | 55.58 |
| TA | 79.69 | 90.08 | 53.04 | 99.96 | 64.43 | 94.23 | 56.33 | 89.66 | 38.73 | 95.10 | 83.97 | 66.55 | 62.69 | 99.96 | 87.12 |
| IBVS | 40.86 | 60.97 | 44.88 | 80.06 | 56.82 | 65.56 | 23.92 | 62.62 | 17.58 | 75.30 | 80.02 | 38.04 | 44.01 | 80.06 | 60.49 |
| SAU | 71.43 | 95.20 | 49.04 | 99.20 | 61.03 | 96.22 | 29.46 | 88.90 | 34.98 | 99.25 | 83.61 | 46.49 | 54.92 | 99.20 | 85.21 |
| SAM | 51.09 | 29.35 | 28.65 | 32.30 | 55.56 | 20.70 | 17.16 | 40.05 | 18.89 | 57.61 | 75.36 | 7.98 | 41.11 | 32.30 | 31.13 |
| PAM | 62.34 | 25.62 | 46.57 | 18.14 | 58.63 | 24.55 | 18.44 | 54.40 | 31.07 | 59.78 | 77.51 | 10.27 | 49.09 | 18.14 | 34.92 |
| LFPM | 74.91 | 5.10 | 49.17 | 2.38 | 60.60 | 5.69 | 42.50 | 11.01 | 32.43 | 16.34 | 82.96 | 3.51 | 57.09 | 2.38 | 8.33 |

BadMerging and LoBAM settings, indicating consistent robustness throughout the entire path.

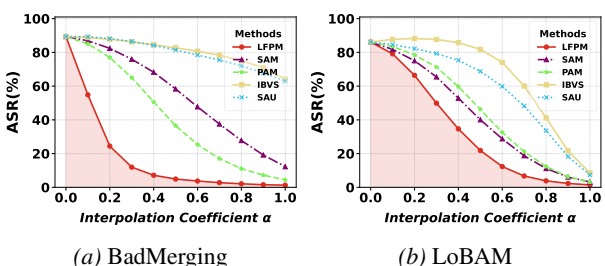

*(a)* BadMerging          *(b)* LoBAM

*Figure 2.* Robustness along the Interpolation Path with Cars196 as the Target Dataset

## 5.3. Robustness against Potential Adaptive Attack

We further evaluate the robustness of LFPM under potential adaptive adversaries. Recall that LFPM is built upon two key principles: (i) **subspace partitioning**, which disentangles adversarial and clean feature components, and (ii) **anti-backdoor task vector optimization**, which performs robust alignment along the feature interpolation path. These design choices implicitly constrain how backdoor signals can be encoded and propagated during model merging.

**Adaptive attack.** An adaptive adversary aware of LFPM may attempt to bypass these mechanisms from two perspectives. First, a *feature coupling attack* enforces strong entanglement between clean and backdoor representations, *e.g.,* by maximizing their cosine similarity on target-class samples, thereby weakening subspace disentanglement. Second, a *feature-interpolation (FI) attack* encourages consistent target-class prediction along the interpolation trajectory between clean and backdoor features, directly counteracting LFPM's path-based optimization. We unify these two strategies into a single adaptive objective, where the loss $\mathcal{L}_{BD}(x_b, y_t)$ jointly models feature coupling and interpolation consistency:

$$\mathcal{L}_{BD}(x_b, y_t) = \mathcal{L}_{CE}\big(h(F_I), y_t\big) + \cos\big(g(x_b; \theta_{adv}), \overline{F}\big). \quad (27)$$

Here, $x_b$ and $x$ denote the backdoor and clean samples, respectively. The model is decoupled as $f = h \circ g$, where $g(\cdot)$ is the feature extractor and $h(\cdot)$ is the linear prediction head. Let $\overline{F} = \mathbb{E}_{x \in \mathcal{B}_t}(g(x; \theta_{pre}))$ be the mean feature of clean target-class samples within a mini-batch $\mathcal{B}_t$, serving as the target-class centroid. The FI term constructs interpolated features $F_I = p \cdot g(x_b; \theta_{adv}) + (1 - p) \cdot g(x; \theta_{pre})$, where the interpolation coefficient is randomly sampled as $p \sim \mathcal{U}[0, 1]$. Notably, unlike the formulation in BadMerging, we use clean sample features from the pretrained model as the interpolation reference, resulting in a stronger and more general attack formulation. Overall, this objective is not specific to LFPM, but serves as a unified and strengthened adaptive attack benchmark for evaluating robustness.

**Results and analysis.** Table 3 reports the performance under adaptive attacks. Compared with standard baselines in Table 1, all methods exhibit varying degrees of degradation under adaptive adversaries. For example, the ASR increases from $63.00\%$ to $99.20\%$ for SAU, from $10.03\%$ to $32.30\%$ for SAM, and from $4.42\%$ to $18.14\%$ for PAM. In contrast, LFPM consistently achieves the lowest ASR, increasing only from $0.49\%$ to $2.38\%$, while maintaining competitive clean accuracy, demonstrating strong robustness against adaptive attack strategies. This robustness stems from the intrinsic design of LFPM. The path-wise optimization via gradient accumulation mechanism, which iteratively aggregates gradients from the backdoor-merged model to correct optimization directions, ensuring consistent alignment with backdoor-relevant signals and enabling persistent backdoor removal even under adaptive attacks.

## 6. Conclusion

In this paper, we propose LFPM, a novel backdoor defense for model merging. Unlike existing methods, LFPM formulates the backdoor robustness of the merged model from a unified feature-space perspective within the CTL framework, enabling effective backdoor mitigation while preserving clean-task performance. Experiments demonstrate that LFPM consistently achieves robustness under both full fine-tuning and PEFT settings. Future work will investigate extending and validating LFPM across diverse scenarios and tasks, aiming to further enhance its generalizability.

## Acknowledgements

This study is supported by the National Key R&D Program of China (Grant No. 2022YFB3102100), National Natural Science Foundation of China (Grant No. U23B2058), Shenzhen Science and Technology Program (Grant No. ZDSYS20210623091809029), and Shenzhen Key Laboratory of Media Security (Grant No. SYSPG20241211174032004).

## Impact Statement

This paper aims to advance the field of machine learning security by studying and improving the backdoor robustness of merged models from a unified feature-space perspective. Our work provides a general framework for mitigating backdoor attacks while preserving clean-task performance across different fine-tuning settings. The potential societal benefit of this work lies in improving the robustness and reliability for model merging, which are increasingly adopted as a cost-effective solution for multi-task deployment. As with most defensive methods, future adaptive attacks may arise, and further research is needed to evaluate effectiveness at larger scales and in more diverse real-world scenarios.

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

# A. Summary of Symbols

The comprehensive list of all notations is summarized in Table 4.

*Table 4.* Glossary of Notations.

| **Spaces and Datasets** | |
| --- | --- |
| $\mathcal{X} \subseteq \mathbb{R}^d$ | Input space |
| $\mathcal{Y} \subseteq \mathbb{R}^c$ | Output (label) space |
| $\Theta \subseteq \mathbb{R}^m$ | Model parameter space |
| $\mathcal{D}_i$ | Dataset for downstream task $\tau_i$ |
| $\mathcal{D}_c$ | Clean dataset (or union of clean task datasets) |
| $\mathcal{D}_b$ | Backdoor dataset (unknown to the defender) |
| $\mathcal{D}_s$ | Shadow dataset used for defense |
| $(x, y)$ | Clean input-label pair |
| $(x_b, y_t)$ | Backdoor sample with target label $y_t$ |
| **Model Architecture** | |
| $f : \mathcal{X} \times \Theta \to \mathcal{Y}$ | Neural network model |
| $f = h \circ g$ | Decomposition into feature extractor $g$ and linear prediction head $h(z) = W^\top z + b$ |
| $\mathcal{M} = \{\mathcal{V}, \mathcal{T}\}$ | CLIP-like model with visual and text encoders |
| $\mathcal{V}$ | Visual encoder (fine-tuned) |
| $\mathcal{T}$ | Text encoder (kept frozen) |
| **Tasks and Task Arithmetic** | |
| $\{\tau_i\}_{i=1}^k$ | $k$ downstream tasks |
| $\boldsymbol{\theta}_0$ | Pre-trained model parameters |
| $\boldsymbol{\theta}_i$ | Parameters fine-tuned on task $\tau_i$ |
| $\Delta\boldsymbol{\theta}_i = \boldsymbol{\theta}_i - \boldsymbol{\theta}_0$ | Task vector for $\tau_i$ |
| $\boldsymbol{\theta}_m = \boldsymbol{\theta}_0 + \sum_{i=1}^k \lambda_i \Delta\boldsymbol{\theta}_i$ | Merged model parameters, with $\lambda_i$ denoting the task merging coefficients |
| **Cross-Task Linearity (CTL)** | |
| $\boldsymbol{\theta}(t) = (1-t)\boldsymbol{\theta}_i + t\boldsymbol{\theta}_j$ | The parameter interpolation path |
| $\delta = \boldsymbol{\theta}_j - \boldsymbol{\theta}_i$ | Parameter displacement vector |
| $\|\Delta(\alpha)\|$ | Feature deviation for CTL, with upper bound $\|\Delta(\alpha)\| \le \lambda_{max}$ |
| $H(\boldsymbol{\theta}(t)) = \bigtriangledown^2 f(\boldsymbol{\theta}(t))$ | The Hessian along the interpolation path |
| $\bigtriangledown_{\boldsymbol{\theta}} \bigtriangledown^2 f(\boldsymbol{\theta}(t))$ | Derivative of the Hessian, with upper bound $\| \bigtriangledown_{\boldsymbol{\theta}} \bigtriangledown^2 f(\boldsymbol{\theta}(t))\| \le J_{max}$ |
| $\|\xi(\alpha)\|$ | Approximation error of the gradient at the interpolated point |
| **Threat Model and Defense** | |
| $\boldsymbol{\theta}_m = \boldsymbol{\theta}_0 + \Delta\boldsymbol{\theta}_c + \Delta\boldsymbol{\theta}_b$ | Backdoored merged model, with components $\Delta\boldsymbol{\theta}_b$ (backdoor) and $\Delta\boldsymbol{\theta}_c$ (clean) |
| $\boldsymbol{\theta}_{k+1} = \boldsymbol{\theta}_0 + \Delta\boldsymbol{\theta}_{k+1}$ | Anti-backdoor task model, where $\Delta\boldsymbol{\theta}_{k+1}$ is the anti-backdoor task vector |
| $\boldsymbol{\theta}'_m = (1-\alpha)\boldsymbol{\theta}_m + \alpha\boldsymbol{\theta}_{k+1}$ | Purified merged model, with $\alpha$ representing the intervention strength |
| **Feature-Space Decomposition** | |
| $P$ | Learnable visual prompt |
| $x' = P \oplus x$ | Adversarial input $x$ |
| $x'' = (P + \Delta P) \oplus x$ | Worst-case prompt-augmented input $x$ |
| $z = g(x)$ | Feature representation of input $x$ |
| $z' = g(x')$ | Feature representation of adversarial input $x'$ |
| $V = S \oplus T$ | Feature space, decomposed into adversarial and clean semantic subspaces $S$ and $T$ |
| $P_s = UU^T$ | Projection matrix onto the adversarial subspace, where $U$ is semi-orthogonal matrix |
| $P_t = I - P_s$ | Projection matrix onto the clean subspace |
| **Optimization and Loss** | |
| $\eta$ | Learning rate |
| $\lambda$ | Scaling factor for gradient accumulation |
| $[\alpha_2, 1.0]$ | Interval used for Loss Path-Integral |

# B. Algorithm

## B.1. Linear Feature Path Minimization

---

**Algorithm 1** Linear Feature Path Minimization (LFPM)

---

1: **Input:** Fine-tuning dataset $\mathcal{D}_s$; Backdoored merged model $f_m(\boldsymbol{\theta}_m) = h_m \circ g_m$; Pre-trained model $f_0(\boldsymbol{\theta}_0) = h_0 \circ g_0$; Merging coefficient $\alpha$; Training hyperparameters $\mathcal{H}_{\text{subspace}}, \mathcal{H}_{\text{anti}}$.

2: **Output:** Purified merged model $f(\boldsymbol{\theta}'_m)$

3: *// Stage I: Adversarial Feature Extraction via via Subspace Partitioning*

4: Invoke Algorithm 2 with $(\mathcal{D}_s, f_m, f_0, \mathcal{H}_{\text{subspace}})$ to obtain:

5:     Adversarial projection matrix $P_s$ and adversarial prompt $P$.

6: *// Stage II: Anti-backdoor Task Vector Optimization*

7: Invoke Algorithm 3 with $(\mathcal{D}_s, f_m, f_0, P, \mathcal{H}_{\text{anti}})$ to obtain:

8:     Anti-backdoor task vector $\Delta\boldsymbol{\theta}_{k+1}$.

9: *// Stage III: Model Purification via Task Arithmetic*

10: Compute purified merged model:
$$\boldsymbol{\theta}'_m = \boldsymbol{\theta}_0 + (1 - \alpha)\Delta\boldsymbol{\theta}_m + \alpha\Delta\boldsymbol{\theta}_{k+1}.$$

11: **Return** purified model $f(\boldsymbol{\theta}'_m)$.

---

## B.2. Adversarial Feature Extraction via Subspace Partitioning

---

**Algorithm 2** Adversarial Feature Extraction via Subspace Partitioning

---

1: **Input:** Fine-tuning dataset $\mathcal{D}_s$; the backdoored merged model $f = h_m \circ g_m$; the pre-trained model $f = h_0 \circ g_0$; outer training epochs $E_{train}$; adversarial optimization steps $E_{adv}$.

2: **Output:** Adversarial projection matrix $P_s$ and Optimized adversarial prompt $P$.

3: **Initialize:** Semi-orthogonal matrix $U \in \mathbb{R}^{n \times r}$, adversarial projection $P_s = UU^\top$, learnable visual prompt $P$;

4: *// Step I: Feature Subspace Partitioning*

5: **for** epoch = 1 **to** $E_{train}$ **do**

6:     **for** each $(x, y) \in \mathcal{D}_s$ **do**

7:         Extract feature from the backdoored merged-model $z = g_m(x)$.

8:         Project features onto adversarial and clean subspaces: $P_s z$ and $(I - P_s)z$.

9:         Update $U$ by minimizing the subspace partitioning objective:
$$\mathcal{L}\big(h_m(P_s z), y\big) - \mathcal{L}\big(h_m((I - P_s)z), y\big) + \mathcal{R}(U), \quad \mathcal{R}(U) = \|U^\top U - I_r\|.$$

10:     **end for**

11: **end for**

12: *// Step II: Prompt-Based Adversarial Feature Mining*

13: **for** step = 1 **to** $E_{adv}$ **do**

14:     **for** each minibatch $\mathcal{B}$ sampled from $\mathcal{D}_s$ **do**

15:         Construct prompt-augmented input $x' = P \oplus x$.

16:         Extract features from clean reference model: $z_c = g_0(x)$ and $z'_c = g_0(x')$.

17:         Extract features from backdoored merged model (pre-trained model): $z = g_m(x)$ and $z' = g_m(x')$.

18:         Estimate batch-wise adversarial feature centroid: $\mu_s = \mathbb{E}_{x \sim \mathcal{B}}[P_s z']$.

19:         Update prompt $P$ by minimizing:
$$\text{sim}\big((I - P_s)z'_c, (I - P_s)z_c\big) - \text{sim}(P_s z', P_s z) + \|P_s z' - \mu_s\|.$$

20:     **end for**

21: **end for**

---

## B.3. Anti-backdoor Task Vector Optimization

---

**Algorithm 3** Anti-backdoor Task Vector Optimization

---

1: **Input:** Fine-tuning dataset $\mathcal{D}_s$; Backdoored merged model $f(\boldsymbol{\theta}_m) = h \circ g$; Pre-trained model $f(\boldsymbol{\theta}_0)$; Adversarial prompt $P$; Prompt radius $r_\rho$; Learning rate $\eta$; Gradient accumulation coefficient $\lambda$; Initialization epochs $E_{init}$; Anti-backdoor optimization epochs $E_{anti}$. Boolean flag `use_path_integral`; Interpolation bound $\alpha_2$; Number of interpolation points $N$; A given model merging operator $\mathcal{M}(\cdot)$;

2: **Output:** Optimized anti-backdoor task vector $\Delta\boldsymbol{\theta}_{k+1}$.

3: *// Step 0: Initial Training of $\boldsymbol{\theta}_{k+1}$*
4: Initialize $\boldsymbol{\theta}_{k+1} = \boldsymbol{\theta}_0$.
5: **for** epoch = 1 **to** $E_{init}$ **do**
6:     **for** each minibatch $\mathcal{B} \subset \mathcal{D}_s$ **do**
7:         $x' \leftarrow P \oplus x$.
8:         Update $\boldsymbol{\theta}_{k+1}$ by minimizing:
$$\mathcal{L}(f(x; \boldsymbol{\theta}_{k+1}), y) + \mathcal{L}(f(x'; \boldsymbol{\theta}_{k+1}), y).$$
9:     **end for**
10: **end for**
11: $\boldsymbol{\theta}_{k+1} \leftarrow \mathcal{M}(\boldsymbol{\theta}_m, \boldsymbol{\theta}_{k+1})$

12: **for** epoch = 1 **to** $E_{anti}$ **do**
13:     **for** each minibatch $\mathcal{B} \subset \mathcal{D}_s$ **do**
14:         *// Step I: Worst-case Prompt Perturbation*
15:         $x'' \leftarrow (P + \Delta P) \oplus x$.
16:         **if** `use_path_integral` **then**
17:             Compute Path-Integral Loss:
$$\mathcal{L}(x''; \boldsymbol{\theta}_m, \boldsymbol{\theta}_{k+1}) = \frac{1}{N} \sum_{i=i_2}^{N} \mathcal{L}\big(h\big((1 - t_i)g(x'; \boldsymbol{\theta}_m) + t_i g(x'; \boldsymbol{\theta}_{k+1})\big); y\big)$$
18:         **else**
19:             Compute standard feature-space interpolation loss:
$$\mathcal{L}(x''; \boldsymbol{\theta}_m, \boldsymbol{\theta}_{k+1}) = \mathcal{L}\big(h\big((1 - \alpha)g(x''; \boldsymbol{\theta}_m) + \alpha g(x''; \boldsymbol{\theta}_{k+1})\big), y\big).$$
20:         **end if**
21:         Update $\Delta P$ by maximizing $\mathcal{L}(x''; \boldsymbol{\theta}_m, \boldsymbol{\theta}_{k+1})$, subject to $|\Delta P| \leq r_\rho$.

22:         *// Step II: Anti-backdoor Task Optimization via Gradient Accumulation*
23:         Recompute adversarial inputs $x'' = (P + \Delta P) \oplus x$ using worst-case $\Delta P$.
24:         **if** `use_path_integral` **then**
25:             Recompute Path-Integral Loss:
$$\mathcal{L}(x''; \boldsymbol{\theta}_m, \boldsymbol{\theta}_{k+1}) = \frac{1}{N} \sum_{i=i_2}^{N} \mathcal{L}\big(h\big((1 - t_i)g(x'; \boldsymbol{\theta}_m) + t_i g(x'; \boldsymbol{\theta}_{k+1})\big); y\big)$$
26:         **else**
27:             Recompute standard feature-space interpolation loss:
$$\mathcal{L}(x''; \boldsymbol{\theta}_m, \boldsymbol{\theta}_{k+1}) = \mathcal{L}\big(h\big((1 - \alpha)g(x''; \boldsymbol{\theta}_m) + \alpha g(x''; \boldsymbol{\theta}_{k+1})\big), y\big).$$
28:         **end if**
29:         Compute total loss:
$$\mathcal{L} \leftarrow \mathcal{L}(f(x; \boldsymbol{\theta}_{k+1}), y) + \mathcal{L}(x''; \boldsymbol{\theta}_m, \boldsymbol{\theta}_{k+1}).$$
30:         Update anti-backdoor model:
$$\boldsymbol{\theta}_{k+1} \leftarrow \boldsymbol{\theta}_{k+1} - \eta\big(\nabla_{\boldsymbol{\theta}_{k+1}}\mathcal{L} + \lambda\nabla_{\boldsymbol{\theta}_m}\mathcal{L}\big).$$
31:     **end for**
32: **end for**
33: **Return** Anti-backdoor task vector: $\Delta\boldsymbol{\theta}_{k+1} = \boldsymbol{\theta}_{k+1} - \boldsymbol{\theta}_0$.

---

## B.4. Curvature Evaluation via Hessian–Vector Products (HVPs)

---

**Algorithm 4** Curvature Evaluation via Hessian–Vector Products (HVPs)

---

1: **Input:** Backdoored merged model $\theta_m$; Anti-backdoor model $\theta_{k+1}$; Number of mini-batches $N$; Adversarial dataset $\mathcal{D}_{\mathrm{adv}}$; Finite-difference step $\epsilon$.
2: **Output:** Average quadratic form $\overline{\mathcal{C}}(\theta; \delta)$ and average effective curvature $\overline{\lambda}_{\mathrm{eff}}$.

3: **Initialize:** $\mathcal{C}_{\mathrm{sum}} \leftarrow 0, \quad \lambda_{\mathrm{sum}} \leftarrow 0$.

4: *// Step I: Compute Parameter Difference*
5: $\delta \leftarrow \theta_{k+1} - \theta_m$.
6: **if** normalize direction **then**
7: $\quad \delta \leftarrow \delta / \|\delta\|$.
8: **end if**

9: **for** $b = 1$ **to** $N$ **do**
10: $\quad$ Sample mini-batch $\mathcal{B}_b = \{(x_i', y_i)\}$ from $\mathcal{D}_{\mathrm{adv}}$.
11: $\quad$ *// Step II: Gradient at Base Parameters*
12: $\quad$ Compute loss $\mathcal{L}(\theta_{k+1}, \mathcal{B}_b)$.
13: $\quad$ Compute gradient $g_0 \leftarrow \nabla_\theta \mathcal{L}(\theta_{k+1})$.
14: $\quad$ *// Step III: Gradient at Perturbed Parameters*
15: $\quad$ $\theta_\epsilon \leftarrow \theta_{k+1} + \epsilon \delta$.
16: $\quad$ Compute loss $\mathcal{L}(\theta_\epsilon, \mathcal{B}_b)$.
17: $\quad$ Compute gradient $g_\epsilon \leftarrow \nabla_\theta \mathcal{L}(\theta_\epsilon)$.
18: $\quad$ *// Step IV: Estimate HVP*
19: $\quad$ $H_\theta \delta \leftarrow (g_\epsilon - g_0)/\epsilon$.
20: $\quad$ *// Step V: Compute Curvature Metrics*
21: $\quad$ $\mathcal{C}_b \leftarrow \langle \delta, H_\theta \delta \rangle$.
22: $\quad$ $\lambda_b \leftarrow \mathcal{C}_b / \|\delta\|^2$.
23: $\quad$ $\mathcal{C}_{\mathrm{sum}} \leftarrow \mathcal{C}_{\mathrm{sum}} + \mathcal{C}_b$.
24: $\quad$ $\lambda_{\mathrm{sum}} \leftarrow \lambda_{\mathrm{sum}} + \lambda_b$.
25: **end for**

26: **Step 6: Average over Mini-batches**
27: $\overline{\mathcal{C}}(\theta; \delta) \leftarrow \mathcal{C}_{\mathrm{sum}}/N$.
28: $\overline{\lambda}_{\mathrm{eff}} \leftarrow \lambda_{\mathrm{sum}}/N$.

29: **Return** $\overline{\mathcal{C}}(\theta; \delta), \overline{\lambda}_{\mathrm{eff}}$.

---

# C. Proofs

## C.1. Proof of Theorem 3.1

For convenience, we restate Theorem 3.1.

**Theorem C.1 (Deviation Bound for CTL).** *Let $f : \mathbb{R}^d \to \mathbb{R}^c$ be a twice continuously differentiable function defined on an open convex set $\Theta \subseteq \mathbb{R}^m$. Consider two fine-tuned models $\boldsymbol{\theta}_i, \boldsymbol{\theta}_j \in \Theta$ originating from the same pretrained checkpoint, and define the linear interpolation path*

$$\boldsymbol{\theta}(t) = (1-t)\boldsymbol{\theta}_i + t\boldsymbol{\theta}_j = \boldsymbol{\theta}_i + t\delta, \tag{28}$$

*where $\delta := \boldsymbol{\theta}_j - \boldsymbol{\theta}_i$ and $t \in [0, 1]$.*

*For $\forall \alpha \in (0, 1)$, define the feature deviation for CTL as $\Delta(\alpha) := f(\boldsymbol{\theta}(\alpha)) - (1 - \alpha)f(\boldsymbol{\theta}_i) - \alpha f(\boldsymbol{\theta}_j)$. Then, the norm of $\Delta(\alpha)$ admits the path-integral form*

$$\|\Delta(\alpha)\| = \int_0^1 k_\alpha(t)\delta^T \bigtriangledown^2 f(\boldsymbol{\theta}(t))\delta dt. \tag{29}$$

*where* $k_\alpha(t) = \begin{cases} (1-\alpha)t, 0 \le t \le \alpha, \\ \alpha(1-t), \alpha \le t \le 1. \end{cases}$

*Moreover, the deviation is bounded by*

$$\|\Delta(\alpha)\| \le \|\delta\|^2 \left[ (1-\alpha) \int_0^\alpha tH(t)dt + \alpha \int_\alpha^1 (1-t)H(t)dt \right] \tag{30}$$

*where $H(\theta(t)) = \bigtriangledown^2 f(\theta(t))$ denotes the Hessian along the interpolation path.*

*Proof. Remark.* Although the output of $f$ is vector-valued, for simplicity we treat it as scalar-valued in the analysis; for vector-valued outputs, all results hold element-wise. This convention applies throughout the proof.

Let $f : \mathbb{R}^d \to \mathbb{R}^c$ be a twice continuously differentiable function defined on an open convex set $\Theta \subseteq \mathbb{R}^m$. Consider two fine-tuned models $\boldsymbol{\theta}_i, \boldsymbol{\theta}_j \in \Theta$ originating from the same pretrained checkpoint, and define the linear interpolation path

$$\boldsymbol{\theta}(t) = (1-t)\boldsymbol{\theta}_i + t\boldsymbol{\theta}_j = \boldsymbol{\theta}_i + t\delta, \tag{31}$$

where $\delta := \boldsymbol{\theta}_j - \boldsymbol{\theta}_i$ and $t \in [0, 1]$.

**Step 1: Taylor expansion along the interpolation path.**

By the second-order Taylor expansion with integral remainder, for $\forall t_0, t_1 \in [0, 1]$, we have

$$f(\boldsymbol{\theta}(t_1)) = f(\boldsymbol{\theta}(t_0)) + \bigtriangledown f(\boldsymbol{\theta}(t_0))^T (\boldsymbol{\theta}(t_1) - \boldsymbol{\theta}(t_0)) + \int_{t_0}^{t_1} (t_1 - t)\delta^T \bigtriangledown^2 f(\theta(t))\delta dt. \tag{32}$$

In particular, taking $t_1 = \alpha$, $t_0 = 0$, we obtain:

$$\begin{aligned} f(\boldsymbol{\theta}(\alpha)) &= f(\boldsymbol{\theta}_i) + \bigtriangledown f(\boldsymbol{\theta}_i)^T (\boldsymbol{\theta}(\alpha) - \boldsymbol{\theta}_i) + \int_0^\alpha (\alpha - t)\delta^T \bigtriangledown^2 f(\boldsymbol{\theta}(t))\delta dt \\ &= f(\boldsymbol{\theta}_i) + \alpha \bigtriangledown f(\boldsymbol{\theta}_i)^T \delta + \int_0^\alpha (\alpha - t)\delta^T \bigtriangledown^2 f(\boldsymbol{\theta}(t))\delta dt, \end{aligned} \tag{33}$$

and taking $t_1 = 1$, $t_0 = 0$ gives

$$f(\boldsymbol{\theta}_j) = f(\boldsymbol{\theta}_i) + \bigtriangledown f(\boldsymbol{\theta}_i)^T \delta + \int_0^1 (1-t)\delta^T \bigtriangledown^2 f(\boldsymbol{\theta}_t)\delta dt. \tag{34}$$

**Step 2: The deviation for CTL.**

Combining Eqs. 33 and 34, the deviation $\Delta(\alpha)$ from linear interpolation in function space, can be expressed as

$$\begin{aligned} \|\Delta(\alpha)\| &= \|f(\boldsymbol{\theta}(\alpha)) - [(1-\alpha)f(\boldsymbol{\theta}_i) + \alpha f(\boldsymbol{\theta}_j)]\| \\ &= \|f(\boldsymbol{\theta}_i) + \alpha \bigtriangledown f(\boldsymbol{\theta}_i)^T \delta + \int_0^\alpha (\alpha - t)\delta^T \bigtriangledown^2 f(\boldsymbol{\theta}(t))\delta dt \\ &\quad - \left[ (1-\alpha)f(\boldsymbol{\theta}_i) - \alpha \big( f(\boldsymbol{\theta}_i) + \bigtriangledown f(\boldsymbol{\theta}_i)^T \delta + \int_0^1 (1-t)\delta^T \bigtriangledown^2 f(\boldsymbol{\theta}(t))\delta dt \big) \right]\| \\ &= \|\int_0^\alpha (\alpha - t)\delta^T \bigtriangledown^2 f(\boldsymbol{\theta}(t))\delta dt - \alpha \int_0^1 (1-t)\delta^T \bigtriangledown^2 f(\boldsymbol{\theta}(t))\delta dt\| \\ &= \|\int_0^\alpha (\alpha - t)\delta^T \bigtriangledown^2 f(\boldsymbol{\theta}(t))\delta dt - \alpha \int_0^\alpha (1-t)\delta^T \bigtriangledown^2 f(\boldsymbol{\theta}(t))\delta dt - \alpha \int_\alpha^1 (1-t)\delta^T \bigtriangledown^2 f(\boldsymbol{\theta}(t))\delta dt\| \\ &= \|(1-\alpha)\int_0^\alpha t\delta^T \bigtriangledown^2 f(\theta(t))\delta dt - \alpha \big( \int_\alpha^1 (1-t)\delta^T \bigtriangledown^2 f(\theta(t))\delta dt \big)\|. \end{aligned} \tag{35}$$

Defining the weighting kernel $k_\alpha(t)$ as

$$k_\alpha(t) = \begin{cases} (1-\alpha)t, 0 \leq t \leq \alpha, \\ \alpha(1-t), \alpha \leq t \leq 1. \end{cases} \tag{36}$$

Then $\Delta(\alpha)$ admits the path-integral form

$$\|\Delta(\alpha)\| = \left\| \int_0^1 k_\alpha(t)\delta^T \bigtriangledown^2 f(\boldsymbol{\theta}(t))\delta dt \right\|. \tag{37}$$

**Step 3: Deviation bound for CTL under a bounded hessian**

Assuming the Hessian $H(t)$ is uniformly bounded along the path, *i.e.,* there exist constants $0 \leq \lambda_{\max}$ such that

$$\|H(t)\| \leq \lambda_{max} \tag{38}$$

we immediately obtain

$$\int_0^1 k_\alpha(t)\delta^T H(t)\delta dt \leq \lambda_{max} \|\delta\|^2 \int_0^1 k_\alpha(t)dt. \tag{39}$$

Finally, computing the integral of the weighting kernel:

$$\int_0^1 k_\alpha(t)dt = (1-\alpha)\int_0^\alpha tdt + \alpha \int_\alpha^1 (1-t)dt = (1-\alpha)\cdot\frac{\alpha^2}{2} + \alpha \cdot \frac{(1-\alpha)^2}{2} = \frac{\alpha(1-\alpha)}{2}. \tag{40}$$

We arrive at the path-integral bound for the CTL deviation:

$$\int_0^1 k_\alpha(t)\delta^T H(t)\delta dt \leq \frac{\alpha(1-\alpha)\lambda_{max} \|\delta\|^2}{2}. \tag{41}$$

$\square$

## C.2. Proof of Lemma 4.1

For convenience, we restate Lemma 4.1.

**Lemma C.2 (Adversarial Perturbation Orthogonality Principle).** *Let $f = h \circ g$, where $g$ is a feature extractor and $h(z) = W^\top z + b$ is a linear prediction head for a downstream task. Suppose an adversarial input $\boldsymbol{x}' = \boldsymbol{x} + \Delta\boldsymbol{x}$ preserves the clean-task semantics,* i.e.,

$$f(x';\boldsymbol{\theta}'_m) = f(x;\boldsymbol{\theta}_m). \tag{42}$$

*Then, under a first-order approximation of $g$, the induced feature perturbation $\Delta z = g(x') - g(x)$ must satisfy*

$$\Delta z \perp \text{span}(W), \tag{43}$$

*where $\text{span}(W)$ denotes the feature subspace spanned by the clean-task feature vectors.*

*Proof.* We recall that $f = h \circ g$, where $g$ is a feature extractor and $h$ is a linear prediction head for a downstream task, with weight matrix $W$. For an input $x$, we denote its feature representation by $z = g(x)$, so that

$$f(x) = h(z) = W^T z + b. \tag{44}$$

**Step 1: Semantics-preserving perturbation constraint.**

Following Section 3.2, the defender mitigates the backdoor effects of the merged model $\boldsymbol{\theta}_m$ by constructing an anti-backdoor task vector $\boldsymbol{\theta}_{k+1}$, such that:

$$\boldsymbol{\theta}'_m = \boldsymbol{\theta}_0 + (1-\alpha)\Delta\boldsymbol{\theta}_m + \alpha\Delta\boldsymbol{\theta}_{k+1} = \boldsymbol{\theta}_m + \alpha\delta \tag{45}$$

where $\delta = \Delta\boldsymbol{\theta}_{k+1} - \Delta\boldsymbol{\theta}_m$, and $\alpha$ controls the strength of the anti-backdoor intervention.

Revisiting the surrogate optimization objective in Section 3.2, adversarial inputs $\boldsymbol{x}' = \boldsymbol{x} + \Delta\boldsymbol{x}$ are required to satisfy the two conditions. Specifically, (i) $x'$ should activate backdoor-related representations in the backdoored merged model, and (ii) it should preserve clean semantic representations in the purified model. These conditions can be formalized as

$$\begin{cases} f(x'; \boldsymbol{\theta}_m) \approx f(x_b; \boldsymbol{\theta}_m) \\ f(x'; \boldsymbol{\theta}'_m) \approx f(x; \boldsymbol{\theta}'_m). \end{cases} \tag{46}$$

Following the standard backdoor definition, where the backdoored merged model $\boldsymbol{\theta}_m$ preserves predictions on clean inputs $x$ and thus behaves consistently with the purified model $\boldsymbol{\theta}'_m$. Based on this property, we relax the clean-feature constraint into a differentiable objective by minimizing the output discrepancy.

$$\min_{\Delta x} \left\| f(x'; \boldsymbol{\theta}'_m) - f(x; \boldsymbol{\theta}_m) \right\| \tag{47}$$

**Step 2: Second-order input-parameter expansion and First-order feature-space approximation.**

We futher expand $f(x'; \boldsymbol{\theta}'_m)$ around $(x, \boldsymbol{\theta}_m)$ using a second-order Taylor expansion with respect to both the input $x$ and the parameters $\boldsymbol{\theta}_m$, we have

$$\begin{aligned} f(x'; \boldsymbol{\theta}'_m) &= f(x + \Delta x; \boldsymbol{\theta}_m + \alpha\delta) \\ &= f(x; \boldsymbol{\theta}_m) + \Delta x^T \bigtriangledown_x f(x; \boldsymbol{\theta}_m) + \alpha * \delta^T \bigtriangledown_{\boldsymbol{\theta}} f(x; \boldsymbol{\theta}_m) \\ &+ \frac{1}{2}\Delta x^T \bigtriangledown_x^2 f(x; \boldsymbol{\theta}_m)\Delta x + \frac{\alpha^2}{2}\delta^T \bigtriangledown_\theta^2 f(x; \boldsymbol{\theta}_m)\delta + \alpha\Delta x^T \bigtriangledown_{x\theta}^2 f(x; \boldsymbol{\theta}_m)\delta + R_3(\delta). \end{aligned} \tag{48}$$

where $R_3(\delta)$ collects higher-order remainder terms.

Focusing only on the terms involving $\Delta x$, Eq. (47) reduces to

$$\min_{\Delta x} \|T(\Delta x, \delta)\| = \left\| \Delta x^T \bigtriangledown_x f(x; \boldsymbol{\theta}_m) + \frac{1}{2}\Delta x^T \bigtriangledown_x^2 f(x; \boldsymbol{\theta}_m)\Delta x + \alpha\Delta x^T \bigtriangledown_{x\theta}^2 f(x; \boldsymbol{\theta}_m)\delta \right\| \tag{49}$$

Since $f = h \circ g$, the feature representation is given by $z = g(x)$ and the model output can be written as $f(x) = h(g(x))$. Under a first-order approximation of the feature extractor $g(x)$ around input $x$, the induced adversarial feature $\Delta z$ by $x'$ satisfies

$$\Delta z = g(x + \Delta x) - g(x) = J_g(x)\Delta x + R_2(\|\Delta x\|^2) \tag{50}$$

where $J_g(x)$ denotes the the Jacobian of $g(\boldsymbol{x})$ at $\boldsymbol{x}$.

Consequently, under the first-order approximation $\Delta z \approx J_g(x)\Delta x$ and neglecting higher-order remainder terms, the input perturbation $\Delta x$ can be reparameterized in the feature space. Applying the chain rule, the objective in Eq. (49) can be expressed as

$$T(\Delta z, \delta) = \Delta z^T \bigtriangledown_z h(z) + \Delta z^T H_{zz}\Delta z + \alpha\Delta z^T H_{z\theta}\delta \tag{51}$$

where $H_{zz} = \bigtriangledown_z^2 h(z)$ and $H_{z\theta} = \bigtriangledown_{z\theta}^2 h(z)$.

**Step 3: Output variation under linear prediction head.**

For a linear prediction head $h(z) = W^\top z + b$, we have

$$\bigtriangledown_z h(z) = W, \qquad H_{zz} = 0, \qquad H_{z\theta} = 0 \tag{52}$$

Therefore, the objective reduces to

$$T(\Delta z, \delta) = \Delta z^T W \tag{53}$$

and the minimization problem becomes

$$\min_{\Delta z} \left\| \Delta z^T W \right\| \tag{54}$$

**Step 4: Orthogonality conclusion.**

Since $W$ spans the feature subspace associated with the clean-task predictions, for non-trivial perturbations $\Delta z \neq 0$, the minimum is achieved when $\Delta z$ lies in the null space of $W^\top$, *i.e.,*

$$\Delta z \perp \text{span}(W) \tag{55}$$

This indicates that, to preserve clean-task semantics under adversarial perturbations, the induced feature perturbation must be orthogonal to the clean-task feature subspace. □

### C.3. Proof of Mapping Feature Perturbation to Prompt Perturbation

In Section 4.2, we show that under the Cross-Task Linearity (CTL) condition (Eq. 1), the original SAM-like optimization along the parameter interpolation path can be equivalently reformulated as robust optimization in feature space:

$$\min_{\Delta\boldsymbol{\theta}_{k+1}} \max_{|\Delta z| \leq \rho} \mathcal{L}\big(h\big(\alpha g(x^{'};\boldsymbol{\theta}_m) + (1-\alpha)g(x^{'};\boldsymbol{\theta}_{k+1})\big) + \Delta z; y\big) \tag{56}$$

where $\alpha g(x^{'};\boldsymbol{\theta}_m) + (1-\alpha)g(x^{'};\boldsymbol{\theta}_{k+1})$ denotes the interpolation of the feature outputs from the feature extractor $g$, and $\Delta z \in \{v : ||v|| \leq \rho\}$, which represents the allowable range of feature-space perturbations.

**Step 1: Feature representation under a prompt.**

Since $x^{'} = P \oplus x$, we denote the corresponding feature representation $z$ as

$$z = g(P \oplus x; \boldsymbol{\theta}) \tag{57}$$

For an initial prompt $P_0$, the feature deviation by induced the prompt-space perturbation $\Delta P$ is

$$\Delta z = g((P_0 + \Delta P) \oplus x; \boldsymbol{\theta}) - g(P \oplus x; \boldsymbol{\theta}). \tag{58}$$

**Step 2: Bounding prompt-space perturbations.**

Applying a first-order Taylor expansion of $g$ with respect to the prompt around $P_0$, we obtain

$$g((P_0 + \Delta P) \oplus x; \boldsymbol{\theta}) \approx g(P_0 \oplus x; \boldsymbol{\theta}) + J_P \Delta P, \tag{59}$$

where $J_P = \frac{\partial z(P)}{\partial P}\big|_{P=P_0}$ denotes the Jacobian matrix of the feature representation $z$ with respect to the learnable prompt $P$. This yields the linear relation:

$$\Delta z \approx J_P \Delta P \tag{60}$$

From Eq. 56, the feature deviations induced by CTL satisfy $\|\Delta z\| \leq \rho$, Consequently, combining this with the linear approximation in Eq. 60, the corresponding prompt-space perturbation $\Delta P$ is bounded as

$$\|\Delta P\| \leq r_\rho, \quad \text{with} \quad r_\rho \approx \frac{\rho}{|J_P|_2}. \tag{61}$$

**Step 3: Reformulating the optimization in prompt space.**

Thus, Eq. 56 can be equivalently reformulated as

$$\min_{\Delta\boldsymbol{\theta}_{k+1}} \max_{|\Delta P| \leq r_\rho} \mathcal{L}[h\big(\alpha g(x^{''};\boldsymbol{\theta}_m) + (1-\alpha)g(x^{''};\boldsymbol{\theta}_{k+1})\big); y] \tag{62}$$

where $x^{''} = (P + \Delta P) \oplus x$ and the prompt perturbation $\Delta P$ is bounded as $r_\rho \leq \frac{\rho}{\|J_P\|_2}$, and $J_P = \frac{\partial z(P)}{\partial P}\big|_{P=P_0}$ denotes the Jacobian matrix of $z(P)$ with respect to the learnable prompt $P$.

### C.4. Proof of Corollary 4.2

To approximate the gradient of an interpolated model $\boldsymbol{\theta}^{'}_m$ using only the endpoint $\boldsymbol{\theta}_m$ and $\boldsymbol{\theta}_{k+1}$, we state the following corollary.

**Corollary C.3 (Gradient Approximation Along Linear Interpolation).** *Under the setting of Theorem 3.1, assume that along the interpolation path* $\boldsymbol{\theta}(t)$,

$$\|\nabla^2 f(\boldsymbol{\theta}(t))\| \leq \lambda_{\max}, \quad \|\nabla_{\boldsymbol{\theta}} \nabla^2 f(\boldsymbol{\theta}(t))\| \leq J_{\max}. \tag{63}$$

*Let* $\boldsymbol{\theta}(\alpha) = (1 - \alpha)\boldsymbol{\theta}_i + \alpha\boldsymbol{\theta}_j$ *for* $\alpha \in (0, 1)$ *and* $\delta = \boldsymbol{\theta}_j - \boldsymbol{\theta}_i$.

*Then, for any* $\alpha \in (0, 1)$,

$$\nabla f(\boldsymbol{\theta}(\alpha)) = (1 - \alpha)\nabla f(\boldsymbol{\theta}_i) + \alpha\nabla f(\boldsymbol{\theta}_j) + \xi(\alpha), \tag{64}$$

*where the approximation error* $\xi(\alpha)$ *satisfies*

$$\|\xi(\alpha)\| \leq 2\alpha(1 - \alpha)\lambda_{\max}\|\delta\| + \frac{\alpha(1 - \alpha)}{2} J_{\max}\|\delta\|^2. \tag{65}$$

*Proof.* Let $f : \mathbb{R}^d \to \mathbb{R}^c$ be a twice continuously differentiable function defined on an open convex set $\Theta \subseteq \mathbb{R}^m$. Consider two fine-tuned models $\boldsymbol{\theta}_i, \boldsymbol{\theta}_j \in \Theta$, and define the linear interpolation path

$$\boldsymbol{\theta}(t) = (1 - t)\boldsymbol{\theta}_i + t\boldsymbol{\theta}_j = \boldsymbol{\theta}_i + t\delta \tag{66}$$

where $\delta := \boldsymbol{\theta}_j - \boldsymbol{\theta}_i$ and $t \in [0, 1]$.

For any $\alpha \in (0, 1)$, consider the linearly interpolated model

$$\boldsymbol{\theta}(\alpha) = (1 - \alpha)\boldsymbol{\theta}_i + \alpha\boldsymbol{\theta}_j \tag{67}$$

**Step 1: Function value decomposition along the interpolation path.**

According to Theorem 3.1, the interpolated model $f(\boldsymbol{\theta}(\alpha))$ can be expressed as

$$\begin{aligned} f(\boldsymbol{\theta}(\alpha)) &= (1 - \alpha)f(\boldsymbol{\theta}_i) + \alpha f(\boldsymbol{\theta}_j) + \Delta Z(\alpha) \\ &= (1 - \alpha)f(\boldsymbol{\theta}_i) + \alpha f(\boldsymbol{\theta}_j) + \int_0^1 k_\alpha(t)\delta^T \nabla^2 f(\boldsymbol{\theta}(t))\delta dt \end{aligned} \tag{68}$$

where $\Delta Z(\alpha) = \int_0^1 k_\alpha(t)\delta^T \nabla^2 f(\boldsymbol{\theta}(t))\delta dt$ with the weighting function

$$k_\alpha(t) = \begin{cases} (1 - \alpha)t, 0 \leq t \leq \alpha, \\ \alpha(1 - t), \alpha \leq t \leq 1. \end{cases} \tag{69}$$

**Step 2: Differentiation with respect to endpoints.**

Differentiating $f(\boldsymbol{\theta}(\alpha))$ with respect to $\boldsymbol{\theta}_i$ and $\boldsymbol{\theta}_j$ yields

$$\begin{cases} \nabla_{\boldsymbol{\theta}_i} f(\boldsymbol{\theta}(\alpha)) = (1 - \alpha) \nabla_{\boldsymbol{\theta}(\alpha)} f(\boldsymbol{\theta}(\alpha)) \\ \\ \nabla_{\boldsymbol{\theta}_j} f(\boldsymbol{\theta}(\alpha)) = \alpha \nabla_{\boldsymbol{\theta}(\alpha)} f(\boldsymbol{\theta}(\alpha)), \end{cases} \tag{70}$$

where $\boldsymbol{\theta}(\alpha)$ denotes the interpolated parameter vector.

Similarly, differentiating the right-hand side of Eq. 68 yields

$$\begin{cases} \nabla_{\theta_i} f(\boldsymbol{\theta}(\alpha)) = (1 - \alpha) \nabla_{\theta_i} f(\boldsymbol{\theta}_i) + \nabla_{\theta_i} \Delta Z(\alpha) \\ \\ \nabla_{\theta_j} f(\boldsymbol{\theta}(\alpha)) = \alpha \nabla_{\theta_j} f(\boldsymbol{\theta}_j) + \nabla_{\theta_j} \Delta Z(\alpha). \end{cases} \tag{71}$$

**Step 3: Interpolated gradient decomposition.**

Combining Eqs. 70 and 71, we obtain

$$
\begin{aligned}
\nabla_{\boldsymbol{\theta}(\alpha)} f\big(\boldsymbol{\theta}(\alpha)\big) &= \nabla_{\boldsymbol{\theta}_i} f\big(\boldsymbol{\theta}(\alpha)\big) + \nabla_{\boldsymbol{\theta}_j} f\big(\boldsymbol{\theta}(\alpha)\big) \\
&= (1-\alpha)\, \nabla_{\boldsymbol{\theta}_i}\, f\big(\boldsymbol{\theta}_i\big) + \alpha\, \nabla_{\boldsymbol{\theta}_j}\, f\big(\boldsymbol{\theta}_j\big) + \nabla_{\boldsymbol{\theta}_i}\Delta Z(\alpha) + \nabla_{\boldsymbol{\theta}_j}\Delta Z(\alpha)
\end{aligned}
\tag{72}
$$

**Step 4: Bounding the gradients of the deviation term.**

Recall that the deviation term along the interpolation path is defined as

$$
\Delta Z(\alpha) = \int_0^1 k_\alpha(t)\delta^T H(t)\delta\, dt, \quad \delta := \boldsymbol{\theta}_j - \boldsymbol{\theta}_i, \quad H(t) = \nabla^2 f(\boldsymbol{\theta}(t)).
\tag{73}
$$

where $\boldsymbol{\theta}(t) = \boldsymbol{\theta}_i + \delta$ is the linearly interpolated parameter and $k_\alpha(t)$ is the weighting function.

To compute the gradients of $\Delta z$ with respect to the endpoint $\boldsymbol{\theta}_i$ and $\boldsymbol{\theta}_j$, we apply the chain rule. Note that $\delta$ depends on the endpoints (*i.e.,* $\nabla_{\boldsymbol{\theta}_i}\delta = -I$ and $\nabla_{\boldsymbol{\theta}_j}\delta = I$) and $H(t)$ depends on the interpolated point $\boldsymbol{\theta}(t)$.

The gradient of the quadratic term $\delta^T H(t)\delta$ is given by:

$$
\begin{cases}
\nabla_{\boldsymbol{\theta}_i}(\delta^T H(t)\delta) = -2H(t)\delta + (1-t)(\delta^T \nabla_{\boldsymbol{\theta}_i} H(t)\delta) \\[2mm]
\nabla_{\boldsymbol{\theta}_j}(\delta^T H(t)\delta) = 2H(t)\delta + t(\delta^T \nabla_{\boldsymbol{\theta}_j} H(t)\delta)
\end{cases}
\tag{74}
$$

Integrating along the interpolation path, the gradients of $\Delta z$ with respect to the endpoints can be expressed as

$$
\begin{cases}
\nabla_{\boldsymbol{\theta}_i}\Delta Z(\alpha) = \int_0^1 k_\alpha(t)\Big[ -2H(t)\delta + (1-t)(\delta^T \nabla_{\boldsymbol{\theta}_i} H(t)\delta)\Big]dt \\[2mm]
\nabla_{\boldsymbol{\theta}_j}\Delta Z(\alpha) = \int_0^1 k_\alpha(t)\Big[2H(t)\delta + t(\delta^T \nabla_{\boldsymbol{\theta}_j} H(t)\delta)\Big]dt
\end{cases}
\tag{75}
$$

where $H(t) = \nabla^2 f(\boldsymbol{\theta}(t))$ is the Hessian along the interpolation path, and $\nabla_{\boldsymbol{\theta}} H(t)$ denotes its derivative with respect to the interpolated point $\boldsymbol{\theta}(t)$. Consequently, we have

$$
\begin{cases}
\|\nabla_{\boldsymbol{\theta}_i}\Delta Z(\alpha)\| \le \int_0^1 k_\alpha(t)\Big[2\,\|H(t)\|\,\delta + (1-t)(\delta^T\,\|\nabla_{\boldsymbol{\theta}_i} H(t)\|\,\delta)\Big]dt \\[2mm]
\big\|\nabla_{\boldsymbol{\theta}_j}\Delta Z(\alpha)\big\| \le \int_0^1 k_\alpha(t)\Big[2\,\|H(t)\|\,\delta + t(\delta^T\,\|\nabla_{\boldsymbol{\theta}_j} H(t)\|\,\delta)\Big]dt.
\end{cases}
\tag{76}
$$

**Step 5: Approximation error of interpolated gradient**

Let $\|\xi(\alpha)\| = \big\|\nabla_{\boldsymbol{\theta}(\alpha)} f\big(\boldsymbol{\theta}(\alpha)\big) - (1-\alpha)\, \nabla_{\boldsymbol{\theta}_i}\, f\big(\boldsymbol{\theta}_i\big) - \alpha\, \nabla_{\boldsymbol{\theta}_j}\, f\big(\boldsymbol{\theta}_j\big)\big\|$, combining Eq. 72 and applying the triangle inequality, we obtain

$$
\begin{aligned}
\|\xi(\alpha)\| &\le \|\nabla_{\boldsymbol{\theta}_i}\Delta Z(\alpha)\| + \big\|\nabla_{\boldsymbol{\theta}_j}\Delta Z(\alpha)\big\| \\
&\le \int_0^1 4k_\alpha(t)\,\|H(t)\|\,\delta + (1-t)k_\alpha(t)\delta^T\,\|\nabla_{\boldsymbol{\theta}_i} H(\boldsymbol{\theta}(t))\|\,\delta + tk_\alpha(t)\delta^T\,\|\nabla_{\boldsymbol{\theta}_j} H(\boldsymbol{\theta}(t))\|\,\delta\, dt
\end{aligned}
\tag{77}
$$

Assume that the Hessian derivative is uniformly bounded along the interpolation path:

$$
\|\nabla_{\boldsymbol{\theta}} H(t)\| \le J_{\max}, \quad \forall t \in [0,1].
\tag{78}
$$

Combined with the constraint from Theorem 3.1:

$$
\|H(t)\| \le \lambda_{max}
\tag{79}
$$

Since $\int_0^1 k_\alpha(t)dt = \frac{\alpha(1-\alpha)}{2}$, we obtain the error bound of gradient :

$$
\|\xi(\alpha)\| \le 2\alpha(1-\alpha)\lambda_{max}\,\|\delta\| + \frac{\alpha(1-\alpha)}{2}J_{\max}\,\|\delta\|^2.
\tag{80}
$$

$\square$

# D. Additional Implementation Details

### D.1. Experiment Settings

For all experiments, we adopt CLIP-ViT-B/32 as the default pre-trained backbone and fine-tune task-specific models on 11 benchmark datasets (Kornblith et al., 2019): CIFAR100, Cars196, SUN397, EuroSAT, GTSRB, Pets, CIFAR10, SVHN, STL-10, Food-101, and RESISC45. All experiments are conducted in multi-task learning scenarios. To evaluate the robustness of the merged model, we consider three randomly shuffled task sequences, each containing six tasks, with the first designated as the adversary task and the second as the target task. A complete summary of all task sequences is provided in Table 5. Task-specific models are obtained under both full fine-tuning and parameter-efficient fine-tuning (PEFT) settings.

*Table 5.* Task sequence used for model merging in multi-task learning. $*$ denotes the default task sequence.

| Task Order | Task Sequence |
|:---:|:---:|
| Task-1$^*$ | CIFAR-100 $\rightarrow$ Cars196 $\rightarrow$ SUN397 $\rightarrow$ EuroSAT $\rightarrow$ GTSRB $\rightarrow$ Oxford-IIIT Pet |
| Task-2 | CIFAR-10 $\rightarrow$ Cars196 $\rightarrow$ SVHN $\rightarrow$ SUN397 $\rightarrow$ STL-10 $\rightarrow$ Oxford-IIIT Pet |
| Task-3 | SVHN $\rightarrow$ CIFAR-10 $\rightarrow$ Food-101 $\rightarrow$ Oxford-IIIT Pet $\rightarrow$ STL-10 $\rightarrow$ RESISC45 |

All task-specific models are trained with the AdamW optimizer for 5 epochs using a batch size of 128. The optimizer is configured with a learning rate of $1.0 * 10^{-5}$, momentum coefficients betas $= (0.9, 0.999)$, a numerical stability term eps $= 1.0 * 10^{-8}$, and a weight decay of 0.001. For LoRA-based PEFT, a higher learning rate of $4 \times 10^{-5}$ is used.

### D.2. Attack Settings

To account for adversaries with varying resource budgets, we consider two representative backdoor attacks for model merging: BadMerging (Zhang et al., 2024) under full fine-tuning and LoBAM (Yin et al., 2024) under parameter-efficient fine-tuning (PEFT) with LoRA. In all attack settings, the adversary injects a backdoor via adversary task vector, such that the triggered samples from the target task are misclassified into an attacker-specified target class. By default, we set class 1 as the target for all target tasks; for example, in the task sequences reported in Table 5, this corresponds to "Acura RL Sedan 2012" for Cars196 and "automobile" for CIFAR10.

**BadMerging** (Zhang et al., 2024) is the first study to investigate backdoor attacks specifically designed for model merging. It introduces a feature-interpolation-based loss to ensure attack effectiveness across a wide range of merging coefficients. Following the official implementation, we sweep the interpolation coefficient $\lambda \in [0.2, 1.0]$.

**LoBAM** (Yin et al., 2024) targets low-resource adversarial settings by fine-tuning pre-trained models with LoRA. To compensate for the reduced attack effectiveness under PEFT, LoBAM amplifies the backdoor task vector, defined as the parameter difference between the backdoored and clean models, thereby enabling effective backdoor implantation for model merging. Following prior work, we set the LoRA rank to $r = 8$ and use an amplification factor of $\lambda = 4.0$. Notably, in the LoBAM setting, all defense methods adopt LoRA-based fine-tuning and are evaluated under the same LoRA configuration to ensure experimental fairness.

### D.3. Defense Settings

We evaluate representative backdoor defense strategies from complementary perspectives. Specifically, these methods include IBVS (Pawlak et al., 2025), which mitigates backdoors through task arithmetic in the parameter space, and SAU (Wei et al., 2023b), a state-of-the-art backdoor defense based on adversarial unlearning. In addition, we consider two sharpness-aware defenses, namely SAM (Foret et al., 2020) and PAM (Min et al., 2024), which enhance robustness by minimizing loss sharpness through weight perturbations during optimization.

**IBVS.** IBVS (Pawlak et al., 2025) is motivated by the insight that backdoor triggers often induce shared and generalizable structure in neural networks, *i.e.*, different triggers tend to activate similar model structures. Leveraging this property, IBVS performs backdoor vector subtraction to mitigate backdoor effect without requiring access to the adversary's dataset, labels, target class, or trigger. Specifically, IBVS applies a fixed trigger to an auxiliary dataset to derive a backdoor task vector and subtracts this vector from the backdoored merged model using task arithmetic. To compensate for potential degradation in clean-task performance, a corresponding clean task vector is incorporated.

In our experiments, we follow the standard protocol and use a fixed white-square trigger on ImageNet-1K (Deng et al., 2009) to obtain the backdoor task vector $V_b$ and clean task vector $V_c$. Given a merged model update $\Delta\theta_m$, the refined update is computed as

$$\Delta\theta_m^{'} = \Delta\theta_m - \lambda(V_b - V_c). \tag{81}$$

After hyperparameter tuning, we set $\lambda = 2.0$, which provides a favorable balance between backdoor mitigation and clean performance preservation.

**SAU.** SAU (Wei et al., 2023b) is a state-of-the-art backdoor defense based on adversarial training, which mitigates backdoors by leveraging shared adversarial examples (SAEs) between the backdoored and purified models, while preserving clean-task performance. While both SAU and LFPM aim to suppress backdoor behaviors without degrading clean-task performance, their mechanisms are fundamentally different: SAU operates in an input-level min-max adversarial training framework, whereas LFPM enhances anti-backdoor task vectors via feature-space interpolation combined with adversarial feature augmentation. In our experiments, SAU is implemented following the open-source BackdoorBench framework (Wu et al., 2025).

**SAM and PAM.** SAM (Foret et al., 2020) and PAM (Min et al., 2024) improve model generalization and robustness by explicitly minimizing loss sharpness through worst-case parameter perturbations in the high-dimensional weight space. Specifically, SAM minimizes both loss value and loss sharpness by computing gradients at worst-case points within a local neighborhood of the current parameters via weight perturbations. PAM follows a similar principle but performs gradient updates at linearly interpolated points between the initial and current model states. At each iteration, both methods require a full optimization cycle in the high-dimensional parameter space, including saving the current parameters, applying weight perturbations, performing forward and backward passes, and finally restoring the original parameters. This process is carried out in the high-dimensional parameter space. In contrast, LFPM transfers perturbations to a low-dimensional prompt space and enforces smoothness along feature-level task trajectories (*e.g.,* the last layer of the backbone), thereby avoiding the full-cycle parameter perturbation.

To ensure a fair comparison, SAM and PAM are applied using the prompt-augmented adversarial inputs $x' = P \oplus x$ generated during Stage I of LFPM.

**LFPM.** LFPM (Foret et al., 2020) comprises two stages: (I) Adversarial Feature Extraction via Subspace Partitioning and (II) Anti-Backdoor Task Vector Optimization.

In Stage I, LFPM learns a rank-$r$ projection matrix $P_s = UU^T$ to partition the feature subspace and extract adversarial features encoded by learnable visual prompts $P$ (Zhou et al., 2022; Jia et al., 2022). In our experiments, we set $U \in \mathbb{R}^{16 \times 512}$ as a semi-orthogonal matrix and use visual prompts of length 10. Prompt optimization is performed using the AdamW optimizer for 5 epochs with a batch size of 64. The learning rate is set to $1.0 * 10^{-4}$, with momentum coefficients betas $= (0.9, 0.999)$, a numerical stability term eps $= 1.0 * 10^{-8}$, and a weight decay of 0.001. To ensure stable training, gradient clipping is applied at each iteration with a maximum prompt norm of 1.0.

In Stage II, LFPM optimizes an anti-backdoor task vector to ensure robust mitigation along the parameter interpolation path. The optimization incorporates three key components:

- **Prompt-space perturbations.** Prompt perturbations $\Delta P$ are bounded by $|\Delta P| \leq r_\rho$ with $r_\rho = 0.1$, following Eq. 20.

- **Gradient Accumulation.** the accumulation coefficient is set to $\lambda = 0.1$, as defined in Eq. 24.

- **Loss Path-integral Optimization.** The integral is evaluated over $\alpha \in [0.9, 1.0]$ with step size of $\Delta t = 0.01$, as described in Eq. 25.

The optimization is performed on a shadow dataset consisting of 10,000 images randomly sampled from ImageNet-1K (Deng et al., 2009). The anti-backdoor task vector is optimized for 5 epochs. In every optimization step, the above three mechanisms are executed once.

Finally, following Step 10 of Algorithm B.1, LFPM merges the anti-backdoor task vector into the backdoored merged model to obtain the purified merged model, using a merging coefficient of $\alpha = 0.5$ for BadMerging and $\alpha = 0.8$ for LoBAM.

# E. Additional Experiments

## E.1. Generality Across Model Merging Algorithms

To evaluate the generality of LFPM and verify that its effectiveness is not tied to a specific model merging strategy, we apply it across several representative merging algorithms:

 (i) **Simple Average (SA)** (Wortsman et al., 2022), which computes a weighted average of model parameters and serves as a widely used baseline for model merging;

 (ii) **Ties-Merging (TIES)** (Yadav et al., 2023), which mitigates parameter interference by sparsifying task vectors and enforcing sign consensus across models;

(iii) **Della-Merging (DELLA)** (Deep et al., 2024), which reduces negative interference through magnitude-aware adaptive sampling applied to task-vector parameters.

These methods represent a broad spectrum of merging approaches, ranging from simple parameter averaging to interference-aware and sparsity-based techniques. By applying LFPM to each of these algorithms, the results summarized in the table 6.

*Table 6.* Defense Performance against BadMerging under Various Model Merging Strategies (ACC↑ / ASR ↓).

| Task → | CIFAR100 | | Cars196 | | SUN397 | | EuroSAT | | GTSRB | | Pets | | Average | | |
|---|---|---|---|---|---|---|---|---|---|---|---|---|---|---|---|
| Method ↓ | CA | ASR(N) | CA | ASR(T) | CA | ASR(N) | CA | ASR(N) | CA | ASR(N) | CA | ASR(N) | CA | ASR(T) | ASR(N) |
| Individual | 83.56 | 75.18 | 64.05 | 79.31 | 71.53 | 56.37 | 97.83 | 30.61 | 94.27 | 49.46 | 88.11 | 42.60 | 83.22 | 79.31 | 55.58 |
| SA | 62.97 | 67.97 | 58.16 | 96.39 | 64.35 | 75.96 | 45.09 | 57.16 | 33.08 | 94.05 | 86.07 | 54.97 | 58.28 | 96.39 | 70.02 |
| SA + LFPM | 65.37 | 11.41 | 54.66 | 0.92 | 62.99 | 9.31 | 37.31 | 29.98 | 30.69 | 5.47 | 82.33 | 5.47 | 55.55 | 0.92 | 16.24 |
| TIES | 72.60 | 80.54 | 51.37 | 99.68 | 63.50 | 85.18 | 43.14 | 62.07 | 46.88 | 92.23 | 79.14 | 74.27 | 59.43 | 99.68 | 78.87 |
| TIES + LFPM | 69.60 | 9.69 | 52.55 | 1.13 | 62.49 | 9.11 | 37.55 | 28.24 | 35.41 | 22.62 | 81.49 | 5.47 | 56.51 | 1.13 | 15.02 |
| DELLA | 72.26 | 84.60 | 49.91 | 99.83 | 63.32 | 87.10 | 49.72 | 59.81 | 58.70 | 92.84 | 78.03 | 77.70 | 61.99 | 99.83 | 80.41 |
| DELLA + LFPM | 70.63 | 9.56 | 52.60 | 1.16 | 62.39 | 9.52 | 39.66 | 25.59 | 38.01 | 20.87 | 81.33 | 5.23 | 57.43 | 1.16 | 14.15 |

From Table 6, we observe that LFPM consistently mitigates backdoor effects in the merged models, achieving an average ASR(T) of around 1% and an average ASR(N) of approximately 15%. At the same time, clean-task performance is largely preserved, with only minor degradation ( 3-4%) compared to the original merged models. These results demonstrate that LFPM operates independently of the underlying merging strategy and provides robust and consistent backdoor mitigation across diverse model merging settings.

## E.2. Robustness under Multiple Compromised Models

Beyond the setting considered in Section 5.3, we further examine a more challenging and realistic scenario with multiple compromised models within the task sequence. This setting reflects realistic cases where the merged-model developer collects multiple task-specific models from third-party sources, some of which may be malicious and contain backdoors originating from multiple independent adversaries.

Specifically, we consider a six-task sequence that contains two distinct "(adversarial task, target task)" pairs, (CIFAR-100, Cars196) and (SVHN, CIFAR-10), along two clean tasks, SUN397 and Pets. This setup enables a systematic evaluation of our method under compounded backdoor threats. The experimental results are summarized in Table 7.

*Table 7.* Backdoor defense performance under multiple compromised models (ACC↑ / ASR ↓).

| Task → | CIFAR100 | | Cars196 | | SVHN | | CIFAR10 | | SUN397 | | Pets | | Average | | |
|---|---|---|---|---|---|---|---|---|---|---|---|---|---|---|---|
| Method ↓ | CA | ASR(N) | CA | ASR(T) | CA | ASR(N) | CA | ASR(N) | CA | ASR(N) | CA | ASR(N) | CA | ASR(T) | ASR(N) |
| Individual | 83.56 | 75.18 | 64.05 | 79.31 | 96.35 | 87.03 | 96.51 | 10.56 | 71.53 | 56.37 | 88.11 | 42.60 | 83.35 | 44.93 | 65.29 |
| TA | 79.97 | 71.48 | 54.69 | 89.46 | 32.56 | 97.91 | 95.37 | 94.34 | 65.36 | 91.91 | 86.01 | 58.51 | 68.99 | 91.90 | 79.95 |
| IBVS | 39.52 | 59.68 | 44.38 | 66.72 | 10.47 | 98.33 | 69.71 | 19.16 | 56.52 | 64.38 | 80.10 | 35.21 | 50.11 | 42.94 | 47.97 |
| SAU | 55.91 | 79.23 | 45.01 | 58.45 | 14.97 | 99.30 | 84.54 | 96.06 | 58.61 | 78.21 | 77.89 | 37.93 | 56.15 | 77.25 | 73.66 |
| SAM | 47.88 | 41.81 | 35.12 | 13.77 | 19.93 | 98.55 | 79.38 | 24.24 | 55.66 | 23.52 | 71.98 | 10.38 | 51.65 | 19.00 | 43.56 |
| PAM | 56.92 | 28.45 | 44.06 | 10.03 | 19.10 | 96.91 | 82.91 | 19.18 | 58.41 | 32.36 | 78.76 | 11.22 | 56.69 | 14.60 | 42.23 |
| LFPM | 74.05 | 6.72 | 52.70 | 0.68 | 39.01 | 9.85 | 93.59 | 9.84 | 63.45 | 8.41 | 83.45 | 3.59 | 67.70 | 5.26 | 7.14 |

From Table 7, we observe that LFPM consistently achieves the best overall defense performance under multiple compromised models. Specifically, LFPM achieves the lowest ASR(T) on both target tasks (0.68% on Cars196 and 9.84% on CIFAR-10) while preserving the highest average CA (67.70%). Notably, LFPM reduces the average ASR(N) of the merged model from 79.95% to 7.14%, indicating its effectiveness in suppressing cross-task backdoor leakage induced by model merging. Consequently, LFPM effectively prevents triggers designed for target tasks from altering predictions on non-target tasks. Overall, these results demonstrate that LFPM robustly mitigates compounded backdoor threats from multiple compromised models while preserving strong clean-task performance.

### E.3. Robustness under Different Task Combinations

We further investigate the effectiveness of backdoor defense methods under different task sequences. Specifically, we consider additional task orders, referred to as Task Order 2 and Task Order 3, as summarized in Table 5. Following the default setting, the first task in each sequence is treated as the adversary task (marked in **blue**), while the second task serves as the target task (marked in **red**). The corresponding results are reported in Tables 8 and 9 for Task Order 2, and in Tables 10 and 11 for Task Order 3.

*Table 8.* Backdoor Defense Performance under BadMerging for Task Order 2 (ACC↑ / ASR ↓).

| Task → | CIFAR10 | | Cars196 | | SVHN | | SUN397 | | STL10 | | Pets | | Average | | |
|---|---|---|---|---|---|---|---|---|---|---|---|---|---|---|---|
| Method ↓ | CA | ASR(N) | CA | ASR(T) | CA | ASR(N) | CA | ASR(N) | CA | ASR(N) | CA | ASR(N) | CA | ASR(T) | ASR(N) |
| Individual | 95.57 | 67.55 | 64.05 | 79.31 | 95.16 | 38.11 | 71.53 | 56.37 | 97.96 | 2.57 | 88.11 | 42.60 | 85.39 | 79.31 | 41.44 |
| TA | 95.96 | 38.47 | 46.07 | 94.85 | 49.14 | 94.14 | 63.65 | 84.73 | 95.40 | 25.38 | 81.43 | 56.58 | 71.94 | 94.85 | 59.86 |
| IBVS | 73.37 | 31.28 | 44.90 | 69.53 | 11.74 | 97.82 | 56.62 | 65.01 | 92.93 | 7.25 | 80.21 | 35.21 | 59.96 | 69.53 | 47.31 |
| SAU | 91.70 | 66.82 | 43.78 | 85.49 | 25.91 | 98.84 | 59.04 | 74.30 | 93.93 | 0.06 | 78.79 | 45.46 | 65.52 | 85.49 | 57.09 |
| SAM | 86.34 | 10.42 | 35.90 | 20.86 | 24.59 | 95.74 | 55.46 | 23.67 | 93.95 | 2.22 | 72.96 | 10.65 | 61.53 | 20.86 | 28.54 |
| PAM | 89.33 | 9.45 | 42.18 | 30.05 | 34.60 | 86.99 | 59.14 | 31.25 | 93.62 | 3.36 | 77.73 | 14.52 | 66.10 | 30.05 | 29.11 |
| LFPM | 95.36 | 1.27 | 50.21 | 2.44 | 50.32 | 41.12 | 63.90 | 13.80 | 96.28 | 0.75 | 83.59 | 6.70 | 73.27 | 2.44 | 12.72 |

*Table 9.* Backdoor Defense Performance under LoBAM for Task Order 2 (ACC↑ / ASR ↓).

| Task → | CIFAR10 | | Cars196 | | SVHN | | SUN397 | | STL10 | | Pets | | Average | | |
|---|---|---|---|---|---|---|---|---|---|---|---|---|---|---|---|
| Method ↓ | CA | ASR(N) | CA | ASR(T) | CA | ASR(N) | CA | ASR(N) | CA | ASR(N) | CA | ASR(N) | CA | ASR(T) | ASR(N) |
| Individual | 95.61 | 67.88 | 61.65 | 54.33 | 93.29 | 17.67 | 67.93 | 48.38 | 97.32 | 2.51 | 89.56 | 18.28 | 84.22 | 54.33 | 30.94 |
| TA | 84.42 | 37.05 | 47.45 | 92.66 | 26.90 | 97.21 | 57.20 | 90.32 | 95.07 | 13.16 | 81.57 | 36.22 | 65.43 | 92.66 | 54.79 |
| IBVS | 81.37 | 25.38 | 46.73 | 19.87 | 21.17 | 53.86 | 53.52 | 61.73 | 94.28 | 8.70 | 76.99 | 34.50 | 62.34 | 19.87 | 36.83 |
| SAU | 81.62 | 37.89 | 40.25 | 22.98 | 30.03 | 80.45 | 57.20 | 67.68 | 94.45 | 20.73 | 83.21 | 29.19 | 64.46 | 22.98 | 47.18 |
| SAM | 86.84 | 12.24 | 50.00 | 7.22 | 23.67 | 70.48 | 62.21 | 30.50 | 95.96 | 1.78 | 86.12 | 15.72 | 67.46 | 7.22 | 26.14 |
| PAM | 85.59 | 13.21 | 48.24 | 8.36 | 26.35 | 64.33 | 61.36 | 33.42 | 95.46 | 2.17 | 85.55 | 18.20 | 67.09 | 8.36 | 26.26 |
| LFPM | 86.79 | 5.47 | 47.99 | 2.94 | 20.51 | 26.88 | 61.88 | 23.22 | 96.45 | 1.32 | 84.98 | 9.34 | 66.43 | 2.94 | 13.24 |

*Table 10.* Backdoor Defense Performance under BadMerging for Task Order 3 (ACC↑ / ASR ↓).

| Task → | SVHN | | CIFAR10 | | Food101 | | Pets | | STL10 | | RESISC45 | | Average | | |
|---|---|---|---|---|---|---|---|---|---|---|---|---|---|---|---|
| Method ↓ | CA | ASR(N) | CA | ASR(T) | CA | ASR(N) | CA | ASR(N) | CA | ASR(N) | CA | ASR(N) | CA | ASR(T) | ASR(N) |
| Individual | 96.35 | 49.90 | 96.51 | 10.56 | 84.69 | 29.90 | 88.11 | 30.79 | 97.96 | 18.86 | 90.95 | 32.03 | 92.42 | 10.56 | 32.29 |
| TA | 79.19 | 92.07 | 91.26 | 100.00 | 79.40 | 84.91 | 84.38 | 76.53 | 95.33 | 89.43 | 63.06 | 97.55 | 82.10 | 100.00 | 88.09 |
| IBVS | 12.65 | 98.36 | 71.53 | 32.92 | 65.62 | 47.25 | 80.40 | 36.19 | 92.93 | 7.55 | 47.93 | 67.77 | 61.84 | 32.92 | 51.42 |
| SAU | 41.82 | 97.90 | 83.24 | 100.00 | 71.40 | 73.04 | 78.74 | 57.56 | 93.55 | 85.53 | 51.33 | 92.46 | 70.01 | 100.00 | 81.29 |
| SAM | 25.66 | 78.54 | 79.11 | 17.84 | 60.08 | 15.13 | 72.28 | 8.17 | 93.47 | 1.86 | 38.93 | 17.82 | 61.58 | 17.84 | 24.30 |
| PAM | 44.47 | 82.80 | 82.50 | 42.82 | 71.80 | 17.49 | 77.56 | 12.21 | 94.36 | 4.21 | 43.50 | 35.31 | 69.03 | 42.82 | 30.40 |
| LFPM | 71.06 | 5.04 | 92.22 | 10.40 | 80.75 | 4.38 | 83.12 | 4.06 | 96.05 | 0.73 | 61.23 | 10.01 | 80.73 | 10.40 | 4.84 |

Across all defense methods, LFPM consistently achieves the strongest defense performance while incurring negligible degradation in clean accuracy. In particular, for both Task Order 2 and Task Order 3, LFPM attains the lowest ASR(T) and

*Table 11.* Backdoor Defense Performance under LOBAM for Task Order 3 (ACC↑ / ASR ↓).

| Task → | SVHN | | CIFAR10 | | Food101 | | Pets | | STL10 | | RESISC45 | | Average | | |
|---|---|---|---|---|---|---|---|---|---|---|---|---|---|---|---|
| Method ↓ | CA | ASR(N) | CA | ASR(T) | CA | ASR(N) | CA | ASR(N) | CA | ASR(N) | CA | ASR(N) | CA | ASR(T) | ASR(N) |
| Individual | 92.39 | 79.92 | 95.61 | 10.14 | 83.00 | 23.98 | 89.56 | 18.28 | 97.32 | 2.51 | 87.61 | 31.23 | 90.91 | 10.14 | 31.18 |
| TA | 24.13 | 94.48 | 84.55 | 99.81 | 75.52 | 66.79 | 82.55 | 63.75 | 95.28 | 83.91 | 52.73 | 73.26 | 69.12 | 99.81 | 76.43 |
| IBVS | 21.25 | 57.62 | 81.53 | 9.11 | 66.61 | 39.96 | 77.37 | 33.38 | 94.33 | 8.32 | 37.06 | 72.17 | 63.02 | **9.11** | 42.29 |
| SAU | 21.53 | 58.20 | 80.86 | 96.71 | 71.80 | 68.34 | 83.34 | 43.77 | 94.97 | 81.97 | 50.04 | 70.77 | 67.09 | 96.71 | 64.61 |
| SAM | 22.06 | 40.78 | **87.26** | 26.98 | 77.59 | 16.81 | **86.23** | 9.18 | 95.90 | 3.07 | 54.63 | 24.22 | 70.61 | 26.98 | 18.81 |
| PAM | **25.87** | **31.97** | 86.27 | 29.47 | 76.13 | 16.08 | 84.68 | 8.80 | 95.66 | 3.75 | 54.69 | 24.20 | 70.55 | 29.47 | 16.96 |
| LFPM | 19.30 | 32.38 | 86.83 | **10.60** | **77.80** | **13.60** | 86.04 | **6.59** | 95.96 | 1.65 | 58.46 | 17.68 | **70.73** | 10.60 | **14.38** |

ASR(N) for nearly all tasks. The only exception occurs under LoBAM for Task Order 3, where ASR(T) reaches 10.60%, slightly above 9.11% of IBVS. Meanwhile, the average CA remains among the highest in most cases. Compared with the original merged model (TA row), LFPM reduces the average CA by only 1.37% under BadMerging for Task Order 3. In other cases, LFPM preserves or even slightly improves CA. For instance, for Task Order 2, CA increases by 1.33% under BadMerging and by 1.00% under LoBAM.

Together with the analysis in Section 5.3, these findings demonstrate that LFPM consistently strikes a favorable balance between backdoor robustness and model performance, regardless of the specific task sequence.

Furthermore, we analyze the evolution of ASR(T) along the parameter interpolation path under Task Orders 2 and 3, with the results shown in Fig. 3 and 4. Consistent with the observations in Fig. 2, LFPM exhibits the smallest Path-ASR, *i.e.,* the smallest area under the ASR curve. These results demonstrate that LFPM provides persistent robustness across the entire interpolation path, regardless of the specific task sequence.

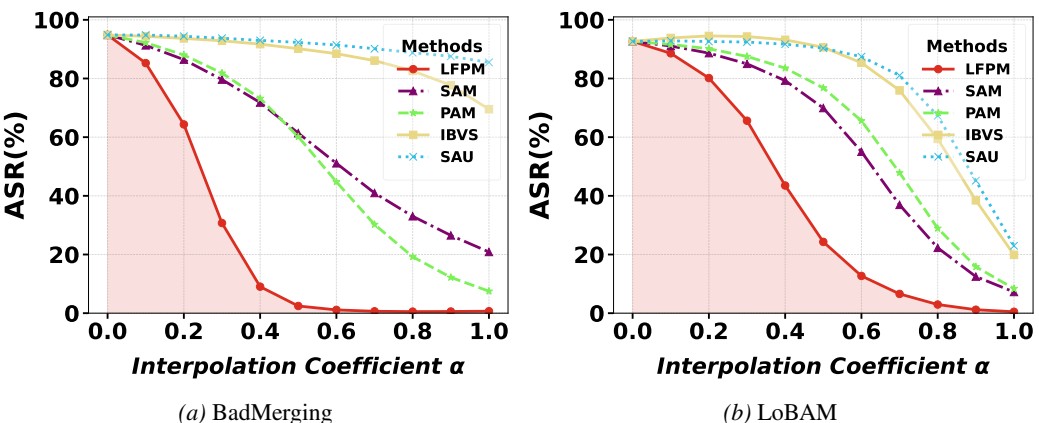

*(a) BadMerging*      *(b) LoBAM*

*Figure 3.* Robustness of the Merged Model along the Interpolation Path (Task Order 2)

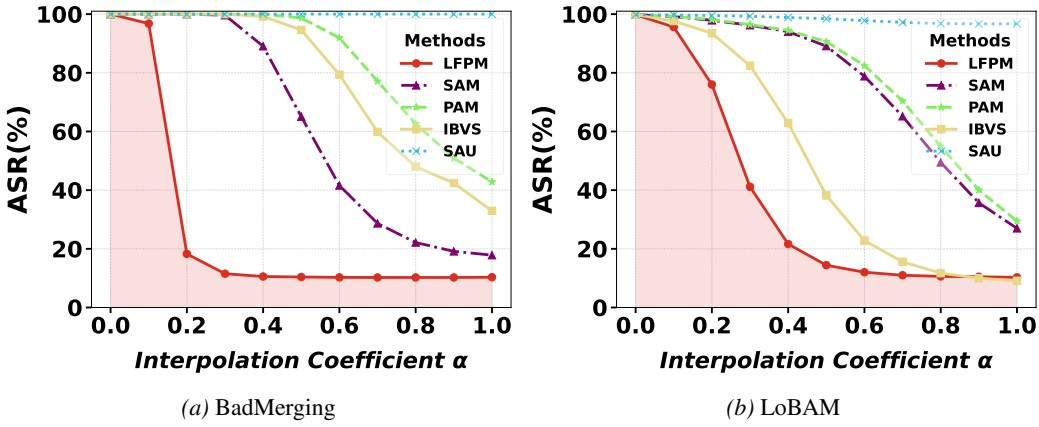

*(a) BadMerging*      *(b) LoBAM*

*Figure 4.* Robustness of the Merged Model along the Interpolation Path (Task Order 3)

## E.4. Scalability to Larger Architectures

To evaluate the scalability of LFPM, we further conduct experiments on a larger backbone, ViT-L/14. The results are reported in Table 12.

*Table 12.* Backdoor Defense Performance under BadMerging on ViT-L/14 (ACC↑ / ASR ↓).

| Task → | CIFAR100 | | Cars196 | | SUN397 | | EuroSAT | | GTSRB | | Pets | | Average | | |
|---|---|---|---|---|---|---|---|---|---|---|---|---|---|---|---|
| Method ↓ | CA | ASR(N) | CA | ASR(T) | CA | ASR(N) | CA | ASR(N) | CA | ASR(N) | CA | ASR(N) | CA | ASR(T) | ASR(N) |
| Individual | 89.21 | 31.57 | 83.82 | 0.38 | 78.61 | 8.36 | 98.59 | 3.61 | 98.15 | 2.16 | 95.20 | 2.45 | 90.59 | 0.38 | 9.63 |
| TA | 91.32 | 17.18 | 61.13 | 85.01 | 63.60 | 63.50 | 32.88 | 76.01 | 50.34 | 83.78 | 90.62 | 38.32 | 64.98 | 85.01 | 55.75 |
| IBVS | 83.78 | 21.01 | 55.22 | 90.26 | 58.41 | 55.18 | 20.66 | 70.44 | **47.45** | 68.58 | 86.56 | 28.12 | 58.68 | 90.26 | 48.66 |
| SAU | 75.74 | 14.66 | 40.51 | 3.25 | 55.21 | 17.63 | 21.25 | 36.18 | 37.15 | 42.97 | 80.07 | 9.51 | 51.65 | 3.25 | 24.19 |
| SAM | 79.89 | 6.29 | 30.66 | 1.72 | 59.59 | 7.02 | 21.05 | 30.68 | 36.23 | 30.88 | 82.58 | 3.65 | 51.66 | 1.72 | 15.70 |
| PAM | 86.01 | 14.40 | 44.93 | 23.57 | 62.77 | 34.83 | 22.05 | 62.87 | 42.95 | 53.27 | 80.92 | 15.37 | 56.60 | 23.57 | 36.14 |
| LFPM | **86.95** | **2.08** | **55.27** | **0.89** | **65.44** | **4.78** | **31.05** | **14.46** | 43.04 | **10.56** | **89.88** | **0.98** | **61.93** | **0.89** | **6.57** |

Overall, LFPM consistently maintains strong defense performance at larger scale, achieving effective backdoor mitigation with an ASR(T) of 0.89% while preserving competitive clean accuracy (average CA of 61.93%). Compared with smaller backbones, LFPM does not exhibit noticeable degradation in either robustness or utility, demonstrating stable performance across model capacities. These results indicate that LFPM is robust to backbone scaling and generalizes well across different model architectures.

## E.5. Empirical Validation of CTL

**CTL Coefficient.** Cross-Task Linearity (CTL) forms the theoretical foundation of our method. To empirically validate this condition, we compute the CTL coefficient along the interpolation path from the backdoored merged model $\theta_m$ to the anti-backdoor task model $\theta_{k+1}$. Following (Zhou et al., 2024), we adopt the CTL coefficient as

$$\text{ceof}(x; \theta_\alpha) = \frac{\|f(x; \theta_\alpha)\| \cos\left[f(x; \theta_\alpha), (1 - \alpha)f(x; \theta_i) + \alpha f(x; \theta_j)\right]}{\|(1 - \alpha)f(x; \theta_i) + \alpha f(x; \theta_j)\|} \tag{82}$$

where $\theta_\alpha = \theta_0 + (1 - \alpha)\Delta\theta_i + \alpha\Delta\theta_j$ denotes the interpolated model. This coefficient measures the alignment in both direction and magnitude between $f(x; \theta_\alpha)$ and $(1 - \alpha)f(x; \theta_i) + \alpha f(x; \theta_j)$, with values closer to 1.0 indicating stronger adherence to the CTL condition (*i.e.*, $f(x; \theta_\alpha) \approx (1 - \alpha)f(x; \theta_i) + \alpha f(x; \theta_j)$ ).

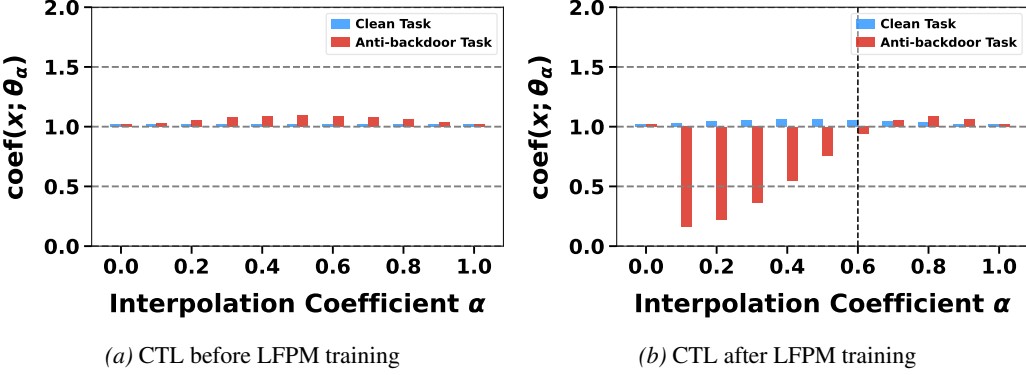

*(a)* CTL before LFPM training      *(b)* CTL after LFPM training

*Figure 5.* Validation of the Cross-Task Linearity Condition

We evaluate the CTL condition for both the initial anti-backdoor model $\theta_{k+1}$ (obtained at Step 11 of Algorithm 3) and the anti-backdoor model after LFPM optimization. As shown in Fig. 5, the CTL condition holds consistently along the entire parameter path for the initial anti-backdoor model. In contrast, after LFPM optimization, CTL is preserved only within a restricted range (*e.g.*, $\alpha \in [0.6, 1.0]$).

Based on the analysis in Theorem 3.3, in order to mitigate the expansion of the deviation for Cross-Task Linearity, we adopt a *Biased merging* strategy. By selecting a value of $\alpha$ closer to an endpoint, we reduce the weighting function $k_\alpha(t)$ and tighten the deviation bound. Accordingly, for the loss path integral in Eq. 25, we set $\alpha_2 = 0.9$. This value is chosen to ensure that the interpolation remains within the valid range of CTL, thus maintaining the theoretical consistency of LFPM.

### E.6. Curvature Analysis

**Motivation.** To better understand the underlying principles of LFPM, we conduct a curvature analysis throughout its dynamic optimization. This analysis aims to confirm that LFPM suppresses rapid growth of CTL deviations during training through its landscape smoothing effect.

As discussed in Section 3.3, the deviation bound for the CTL condition can be quantified as

$$\mathcal{C}(\theta; \delta) = \left\| \delta^\top H(\theta(t)) \delta \right\| \tag{83}$$

where $H(\theta) = \nabla^2 f(\theta(t))$ denotes the Hessian of the neural network evaluated at the interpolated parameters state $(1 - \alpha)\theta_i + \alpha\theta_j$, and $\delta = \theta_j - \theta_i$ represents the difference between the endpoint models.

This quantity measures the upper bound of the feature deviation $\Delta(\alpha)$, *i.e.*, $f(\boldsymbol{\theta}(\alpha)) - (1 - \alpha)f(\boldsymbol{\theta}_i) - \alpha f(\boldsymbol{\theta}_j)$. A small and stable $\mathcal{C}(\theta; \delta)$ indicates strong adherence to CTL.

**HVPs without Explicit Hessian Computation.** Direct computation of $\mathcal{C}(\theta; \delta)$ is intractable in practice, as it requires evaluating the Hessian $H(\theta) = \nabla^2 f(\theta(t))$, which is computationally infeasible for deep neural networks with a large number of parameters. To overcome this challenge, we estimate it efficiently using Hessian-vector products (HVPs) (Dagreou et al., 2024) without explicitly constructing the Hessian. Fellowing Pearlmutter (Pearlmutter, 1994), a Hessian-vector product $\langle \nabla^2 f(\theta), v \rangle$ can be computed as the directional derivative of the gradient along $v$:

$$\nabla^2 f(\theta)\, v = \lim_{\epsilon \to 0} \frac{1}{\epsilon} \left[ \nabla f(\theta + \epsilon v) - \nabla f(\theta) \right] = \nabla_\theta \big( \langle \nabla f(\theta), v \rangle \big). \tag{84}$$

This enables efficient computation via automatic differentiation at a small overhead relative to the standard gradient computation. Using the HVP $H_\theta \delta$, the deviation is then obtained as

$$\mathcal{C}(\theta; \delta) = \langle \delta, H_\theta \delta \rangle. \tag{85}$$

**Curvature Analysis Algorithm.** For each iteration of the anti-backdoor optimization, we analyze curvature along the interpolation path from the backdoored merged model $\theta_m$ to the anti-backdoor model $\theta_{k+1}$ using the adversarial dataset $\mathcal{D}_{\mathrm{adv}}$. The procedure is summarized in Algorithm B.4.

For each epoch, we compute the effective curvature $\lambda_{\mathrm{eff}} = \frac{\delta^T H \delta}{\|\delta\|^2}$, with the results reported in Table 13.

*Table 13.* Evolution of Curvature $\lambda_{\mathrm{eff}}$ throughout Training

| Epoch | 0 | 1 | 2 | 3 | 4 | 5 |
|---|---|---|---|---|---|---|
| $\delta^T H \delta$ | 1.81 | 1.40 | 2.32 | 2.21 | 1.75 | 1.53 |
| $\|\delta\|^2$ | 0.33 | 1.38 | 2.89 | 3.72 | 5.18 | 6.85 |
| $\lambda_{\mathrm{eff}}$ | 5.47 | 1.01 | 0.80 | 0.59 | 0.34 | 0.22 |

From Table 13, we observe that although the parameter distance $\|\delta\|^2$ enlarges nearly 20-fold (6.85/0.33), the CTL deviation remains tightly bounded. This stability is attributed to the smoothing effect of LFPM, which progressively reduces the effective curvature $\lambda_{\mathrm{eff}}$ from 5.47 to 0.22. These results empirically validate our theoretical analysis and provide insight into the role of curvature smoothing in the robustness of LFPM.

### E.7. Computational Cost Comparison

We evaluate the computational efficiency of LFPM by comparing its runtime and GPU memory against representative backdoor defense methods.

*Table 14.* Computational Cost Comparison of Backdoor Defense Methods

| Method | IBVS | SAU | SAM | PAM | LFPM |
|---|---|---|---|---|---|
| Total Time (s) | 13356.6 | 1319.6 | 1085.7 | 845.70 | 1152.5 |
| GPU Peak (MB) | 5079.1 | 8588.6 | 4457.2 | 5125.0 | 10858.8 |

As shown in Table 14, LFPM achieves a favorable trade-off between efficiency and effectiveness. It is slightly less efficient than lightweight SAM-based methods (*e.g.,* SAM and PAM), but significantly more efficient than SAU and substantially faster than task-vector-based approaches such as IBVS, which incur high computational cost due to the requirement of both clean and backdoored models to construct task vectors. Overall, LFPM operates on a single merged model and thus the number of task-specific models does not introduce additional runtime overhead, making its computational cost moderate and scalable compared with existing defense paradigms.

In terms of GPU memory consumption, LFPM requires higher memory than SAM and PAM due to the use of a full merged model during optimization. However, this characteristic is also shared by methods such as SAU. Importantly, such overhead can be effectively mitigated using standard techniques, including mixed-precision training and parameter-efficient fine-tuning (*e.g.,* LoRA), which reduce trainable parameters while keeping most of the model frozen. These techniques are orthogonal to our method and do not affect its applicability or effectiveness. We leave further memory optimization as future work.

### E.8. Efficiency Analysis

Based on the principles of Sharpness-Aware Minimization (SAM), the proposed LFPM algorithm introduces several novel mechanisms: **C1:** Worst-case Perturbations in Prompt Space; **C2:** Gradient Accumulation Approximation; **C3:** Loss Path-Integral. To evaluate the additional time and memory overhead introduced by these mechanisms, we compare LFPM with the standard SAM algorithm (Foret et al., 2020). Specifically, both methods are tested under BadMerging with full fine-tuning setting, conducted on four NVIDIA RTX 3090 GPUs for 5 epochs, as detailed in the experimental configurations in Appendix D.1. The resulting experimental results are summarized in Table 15.

*Table 15.* The Results of Efficiency Analysis for LFPM

| Method | Time per Epoch (s) | Total Time (s) | GPU Peak (MB) |
|---|---|---|---|
| SAM | 121.5 | 610.5 | 4457.2 |
| C1 | 127.9 | 642.0 | 10848.2 |
| C1 + C2 | 128.3 | 645.6 | 10863.1 |
| C1 + C2 + C3 | 136.3 | 677.3 | 10858.8 |

For each method, we report the time cost per training epoch (using randomly selected single-epoch metrics, rather than average values), the total time cost, and the peak memory usage across all GPUs, which serves as an indicator of memory bottlenecks. From the Table 15, we observe that for condition C1 (using only Worst-case Perturbations in Prompt Space), the time cost is comparable to that of the standard SAM (slightly higher than SAM). However, LFPM (C1 + C2 + C3) incurs a slight increase in time cost, specifically an additional time cost of approximately 15 seconds per epoch, resulting in a total increase of 66.8 seconds. This increase is primarily due to the introduction of the Gradient Accumulation Approximation and Loss Path-Integral mechanisms. In terms of memory usage, LFPM requires approximately twice the memory of standard SAM, primarily due to the inclusion of the backdoored merged model as a reference model.

LFPM is designed to maintain consistent robustness along the interpolation path, rather than optimizing for algorithmic efficiency. Given the practical challenges posed by backdoor attacks in model merging, we consider these additional time and memory costs to be justified. In future work, we will focus on improving the efficiency of the LFPM algorithm, such as by employing caching mechanisms to avoid redundant computations and storage of intermediate results.

### E.9. Ablation Studies

In this section, we conduct ablation studies to examine the contributions of individual components within the LFPM framework. As described in Section 4, LFPM consists of two sequential stages: *Adversarial Feature Extraction* and *Anti-backdoor Task Vector Optimization*. The first stage uses **Feature Subspace Partitioning** mechanism to extract adversarial features by disentangling adversarial features from clean semantic features, thereby preserving the model's performance on clean tasks. The second stage incorporates several complementary mechanisms, including **Worst-case Prompt-space Perturbations** to enhance robustness of the merged model, as well as **Gradient Accumulation Approximation** and **Loss Path-Integral** optimization to ensure effective backdoor mitigation along the entire interpolation path.

For clarity, we denote these components as:**(A)** Feature Subspace Partitioning; **(B)** Worst-case perturbations in prompt space; **(C)** Gradient Accumulation Approximation and Loss Path-Integral optimization. Under this notation, the complete LFPM framework corresponds to the combination **A + B + C**. To evaluate the impact of each component, we examine four configurations: **B + C**, **A + C**, **A + B**, and the full method **A + B + C**. The experimental results are summarized in Table 16.

*Table 16.* Results of the Ablation Study on LFPM (ACC↑ / ASR ↓).

| Task → | CIFAR100 | | Cars196 | | SUN397 | | EuroSAT | | GTSRB | | Pets | | Average | | |
|---|---|---|---|---|---|---|---|---|---|---|---|---|---|---|---|
| Method ↓ | CA | ASR(N) | CA | ASR(T) | CA | ASR(N) | CA | ASR(N) | CA | ASR(N) | CA | ASR(N) | CA | ASR(T) | ASR(N) |
| Individual | 83.56 | 75.18 | 64.05 | 79.31 | 71.53 | 56.37 | 97.83 | 30.61 | 94.27 | 49.46 | 88.11 | 42.60 | 83.22 | 79.31 | 55.58 |
| B + C | 65.00 | 14.11 | 54.03 | 3.86 | 61.07 | 16.37 | 38.20 | 30.37 | 30.78 | 34.18 | 82.16 | 6.92 | 55.20 | 3.86 | 20.39 |
| A + C | 70.18 | 13.31 | 54.28 | 1.44 | 63.72 | 16.62 | 36.40 | 36.14 | 34.27 | 30.19 | 84.65 | 9.37 | 57.25 | 1.44 | 20.12 |
| A + B | 70.06 | 28.58 | 55.59 | 39.93 | 65.34 | 39.09 | 54.48 | 42.11 | 39.90 | 57.03 | 84.38 | 18.83 | 61.62 | 39.93 | 37.12 |
| A + B + C | 68.96 | 10.13 | 55.76 | 1.13 | 63.32 | 8.28 | 45.03 | 28.24 | 34.82 | 23.33 | 84.35 | 4.19 | 58.70 | 1.13 | 14.83 |

From Table 16, we observe that the configuration "B + C" achieves a backdoor mitigation performance comparable to that of "A + B + C". However, it incurs a clear degradation in clean accuracy, with the average CA dropping by 3.5% compared to "A + B + C" (55.20% vs. 58.70%). This result highlights the critical role of the Feature Subspace Partitioning mechanism in preserving clean-task performance.

Furthermore, both "A + C" and "A + B" demonstrate a certain degree of backdoor mitigation. Compared with "A + B", the "A + C" configuration achieves substantially lower ASR , reducing ASR(T) from 39.93% to 1.44% and the average ASR(N) from 37.12% to 20.12%, at the cost of a moderate reduction in the average CA ( 57.25% vs 61.62%). These results suggest that both the Worst-case Prompt-space Perturbations mechanism and the optimization strategy based on Gradient Accumulation Approximation and Loss Path-Integral contribute to mitigating backdoor behaviors. Nevertheless, the latter provides stronger backdoor suppression at the expense of clean-task performance.

Overall, only the "A + B + C" configuration, corresponding to the complete LFPM framework, achieves a favorable balance between effective backdoor mitigation and clean-task performance preservation.

