# OpenReview forum: "From Parameters to Feature Space: Task Arithmetic for Backdoor Mitigation in Model Merging"
_ICML.cc/2026/Conference — ICML 2026 regular_

### Official Review · Reviewer_5Hoe · 2026-02-14

**Soundness:** 3
**Presentation:** 3
**Significance:** 2
**Originality:** 3
**Overall Recommendation:** 4
**Confidence:** 2

**Summary:**

This paper proposes Linear Feature Path Minimization as a new backdoor defense in model merging scenarios. Inspired by cross-task linearity, the authors propose to convert the parameter-space robustness to the feature-space robustness. Under this view, the defender constructs an anti-backdoor task vector and merges it with the backdoored merged model to obtain a purified model. Two steps are employed: the defender first mines adversarial features by optimizing the prompt $P$, and then they optimize for the anti-backdoor task vector. Experiments on CLIP-ViT-B-32 across various image datasets show the effectiveness of the proposed defense: It stands out in the BadMergeing and shows competitive performance in the LoBAM.

**Compliance With Llm Reviewing Policy:**

Affirmed.

**Final Justification:**

I incline to the positive side (4->4) after the rebuttal, as the authors gave detailed clarification and complementary evidence.
The authors addressed my concerns regarding the applicability to other model architectures, computational cost with larger models, and the reliability of experimental results. These further reinforced my assessment.

**Key Questions For Authors:**

- What about applying the method to larger models and newer models?  The efficiency analysis in the Appendix reveals that the method is memory-intensive. What about the efficiency aspect when being applied to larger models?
- What about extending the attacks and the proposed defense to LLM or MLLMs?

**Limitations:**

Not discussed. See cons for suggestions.

**Strengths And Weaknesses:**

#### Pros

1. This paper is well organized and grounded in solid theory.

2. New insight: Convert the parameter-space robustness to feature-space robustness. The shift from parameter-space task editing to a feature-space formulation is also a meaningful conceptual contribution.


#### Cons

1. The current evaluation setup is narrow. While the experiments cover various image tasks, the current evaluation only includes CLIP-ViT-B-32. This renders it unclear whether the empirical effectiveness is model-dependent.

2. The random seed variance is not characterized. What about the variance incurred by the random seed? For example, in Table 2, the LFPM only shows analogous performance to SAM. The experimental advantage can be noisy and should be better justified. Repeating the experimental runs and reporting the mean (std) are required.

3. While the key insight is conceptually general, the proposed method seems to be tailored to CLIP-based models. Some tricks are particularly model-dependent, for example, learnable visual prompts. It is uncertain how to extend the method to plain supervised ViTs, CNNs, or other decoder-based models.

---

> ### Author Rebuttal · Authors · 2026-03-31
>
> **Q1: What about applying the method to larger models and newer models? The efficiency analysis in the Appendix reveals that the method is memory-intensive. What about efficiency when applied to larger models?**
>
> **R1:** We thank the reviewer for the important question. We analyze the efficiency of LFPM from three aspects.
>
> * **Computational time comparison.** **As shown in Table 3, LFPM achieves a moderate runtime among existing defenses** . **Notably** , it consists of three stages (Subspace Partitioning, Adversarial Feature Mining, and Anti-backdoor Task Optimization), with an overall runtime of 1152.5s, where the first two stages take 159.3s and 315.8s, respectively. Compared with prior methods, task-vector-based approaches (e.g., IBVS) incur the highest cost due to training both clean and backdoored models. In contrast, SAM-like methods (e.g.,SAM/PAM ) are more efficient, but provide weaker defense performance. LFPM lies in between: achieving a favorable trade-off between performance and efficiency. **Importantly, LFPM operates on a single merged model, so no additional overhead is introduced by task-specific models** .
> * **Scalability to larger models.** LFPM relies mainly on standard forward/backward passes over the merged model, aligning closely with SAM-like optimization. **Thus,  it does not introduce additional model replicas or architecture-dependent modules, its runtime overhead increases smoothly with model scale.**
> * **Memory overhead and mitigation.** LFPM uses the merged model as a reference, leading to higher GPU memory usage than SAM/PAM. However, this is also shared by methods such as SAU. This overhead can be reduced using mixed-precision training or parameter-efficient fine-tuning (e.g., LoRA), which reduces trainable parameters while keeping most weights frozen. **These techniques are orthogonal to LFPM and do not affect its effectiveness** . We leave further memory optimization as future work.
>
> **Table 3: Computational Cost Comparison of Backdoor Defense Methods**
>
> | Method      | IBVS    | SAU    | SAM    | PAM    | LFPM    |
> |------------|--------:|-------:|-------:|-------:|--------:|
> | Total Time (s) | 13356.6 | 1319.6 | 1085.7 | 845.70 | 1152.5 |
> | GPU Peak (MB)  | 5079.1  | 8588.6 | 4457.2 | 5125.0 | 10858.8 |
>
> ---
>
> **Q2: What about extending the attacks and proposed defense to LLMs or MLLMs?**
>
> **R2:** We thank the reviewer for this comment. **LFPM is not tied to CLIP-style models, but built on a model-agnostic feature-space formulation**. Its key components are:
>
> - **Prompt-based adversarial feature extraction:** `uses learnable prompts to encode adversarial perturbations.`
> - **SAM-like optimization on feature interpolation paths:** enforces robustness along feature-space interpolation.
> - **Gradient accumulation from the merged model:** approximates path-wise gradients by accumulating gradients from the merged reference model.`
>
> Importantly, the latter two components operate at the feature/optimization level and are architecture-independent.
>
> **Extension to different architectures.**
>
> - **Prompt-based component.** Visual prompts are one instantiation. For ViTs/CNNs, they can be replaced by input perturbations, feature perturbations, or adapter modules. For LLMs/MLLMs, this can be implemented via soft prompts[1] or multimodal prompts[2].
> - **Optimization components.** SAM-like interpolation-path optimization and gradient accumulation require only a representational space and a differentiable interpolation path, making them transferable across model families.
>
> **Overall**, LFPM is a general feature-space optimization framework and is extensible beyond CLIP-style models.
>
> ---
>
> **Q3: How robust are results to random seed variance? Should mean and std be reported?**
>
> **R3:** Thanks. We repeat experiments five times under the LoBAM attack setting for both SAM and LFPM with different seeds and report mean ± std.
>
> * **For LFPM**: CA: 54.21±0.31, ASR(T): 4.37±1.05, ASR(N): 25.88±0.89.
> * **For SAM**: CA: 51.97±0.66, ASR(T): 3.96±0.94, ASR(N): 23.59±0.97.
>
> Both methods show low variance, indicating stable performance across seeds. The results confirm that LFPM consistently achieves slightly higher CA while showing a comparable trade-off in ASR compared to SAM.
>
> # References
>
> [1] K. Zhou et al., “Learning to prompt for vision-language models,” Int. J. Comput. Vis., vol. 130, no. 9, pp. 2337–2348, 2022.
>
> [2] M. U. Khattak​ et al​, “MaPLe: Multi-modal prompt learning,” in CVPR, 2023, pp. 19113–19122.

---

> > ### Author Rebuttal · Reviewer_5Hoe · 2026-04-01
> >
> > Most of my concerns are solved. But I will appreciate it if experiments with models beyond CLIP-style models can be provided.

---

> > > ### Author Response · Authors · 2026-04-04
> > >
> > > **Q1: Experiments validating beyond CLIP-style models can be provided.**
> > >
> > > **R1:** We thank the reviewer for the positive feedback and the suggestion to evaluate generalization beyond CLIP-style models.
> > >
> > > We additionally evaluate LFPM on a CNN backbone (ResNet-50) to verify its generalization. **Notably**, extending model merging methods to CNNs is less explored, as prior works (e.g., task arithmetic, Ties-Merging) mainly focus on transformer-based models, and their effectiveness on CNNs is often less stable in practice. Since LFPM is originally designed for merged backdoored models, we evaluate it in a task-specific single-model setting, which isolates the core feature-space optimization mechanism from model-merging-specific factors and allows us to directly assess its architecture-agnostic property.
> > > **Therefore, we conduct experiments under a standard supervised image classification setting using a plain CNN backbone (ResNet-50).**
> > >
> > > **Experimental setup.** On the ResNet-50 architecture, we replace the original learned visual prompt with an adapter module to encode adversarial perturbations, while keeping the other components of LFPM unchanged, including SAM-like optimization along feature interpolation paths and the gradient accumulation approximation.
> > >
> > > Specifically, we follow the design of Conv-Adapter [1] and insert adapter modules sequentially into selected residual blocks of ResNet-50. Each Conv-Adapter consists of two convolutional layers with a non-linear activation function in between. The adapter is applied in a residual manner. Let $x$ denote the input feature of the block, and let $h = F(x)$ denote the output of the original residual branch. **The final output of the residual block is given by**:
> > >
> > > $$
> > > y =  x + h + \alpha \cdot \Delta h,
> > > $$
> > >
> > > where $x + h$ is the standard residual output, and $\Delta h = \text{Adapter}(h)$ is the adapter modulation, and $\alpha$ is a scaling factor controlling the strength of the adapter.
> > > **In our implementation, we adopt a front-insertion strategy, i.e., the adapters are inserted into the early residual blocks of the network**. We set the number of adapted blocks to 2 and $\alpha = 0.5$, following the lightweight design principle of Conv-Adapter.
> > >
> > > **Results and Conclusion**. In our experiments, we use CIFAR-100 as the target dataset, with the target class set to 1 under the BadMerging attack.  The backdoored model achieves a CA/ASR of 80.11%/100.00%, and after applying LFPM, **the results improve to 68.66%/7.62%, effectively suppressing the backdoor behavior**. We note that maintaining clean accuracy (CA) on CNN backbones remains an open challenge and warrants further investigation.
> > >
> > > **Overall**, these results demonstrate that LFPM is a general feature-space optimization framework that is not inherently tied to CLIP-style models, and can be extended to both transformer-based and CNN-based architectures.
> > >
> > > # References
> > >
> > > [1] Chen H., et al. Conv-Adapter: Exploring Parameter Efficient Transfer Learning for ConvNets. CVPR 2024, pp. 1551–1561.

---

### Official Review · Reviewer_WhCz · 2026-03-07

**Soundness:** 3
**Presentation:** 3
**Significance:** 3
**Originality:** 2
**Overall Recommendation:** 4
**Confidence:** 2

**Summary:**

The paper proposes Linear Feature Path Minimization (LFPM), a backdoor defense for model merging that moves from parameter-space task arithmetic to a unified feature-space perspective grounded in Cross-Task Linearity (CTL). LFPM learns an anti-backdoor task vector by (i) extracting adversarial features via feature subspace partitioning with learnable visual prompts, and (ii) optimizing a SAM-like robust objective along an interpolation path in feature space, supported by a gradient-accumulation approximation and a loss path-integral mechanism. Experiments on CLIP-based vision tasks show that LFPM reduces attack success rates under both full fine-tuning and PEFT (LoRA) merging while largely preserving clean accuracy, and improves robustness along interpolation paths.

**Compliance With Llm Reviewing Policy:**

Affirmed.

**Final Justification:**

The author's reply solves my concerns, and I keep the original score.

**Key Questions For Authors:**

1. The theoretical analysis (e.g., Theorem 3.1 and the gradient approximation result) appears to ignore the dependence on input x. Are the results intended to hold per-sample or in expectation over the data distribution?
2. How is the Jacobian norm used to determine the prompt perturbation radius estimated in practice? Could the authors provide an ablation showing the sensitivity of LFPM to this parameter?
3. The method assumes a linear prediction head when deriving the orthogonality property of adversarial perturbations. How does LFPM behave when the classifier head includes normalization or other nonlinear components?
4. Have the authors evaluated adaptive attacks that explicitly target the LFPM defense (e.g., attacks optimized against the feature interpolation objective)? If not, how robust do the authors expect LFPM to be under such settings?

**Limitations:**

yes

**Strengths And Weaknesses:**

Strengths

1. The paper introduces a feature-space perspective for backdoor mitigation in model merging under the CTL framework, extending prior work that mainly operates directly in parameter space.
2. The proposed LFPM framework combines feature subspace partitioning with prompt-based adversarial feature mining and a SAM-like optimization along feature interpolation paths, forming a coherent and reasonably well-motivated pipeline.
3. Experiments cover both full fine-tuning attacks and PEFT attacks across multiple tasks and datasets, and the results show consistent reductions in attack success rates while largely preserving clean accuracy.

Weaknesses

1. The orthogonality-based argument for adversarial feature extraction relies on the assumption of a linear prediction head and exact semantic preservation, which may limit the generality of the method when classifiers include normalization or other nonlinear components.
2. The mapping from feature deviation to prompt-space perturbation relies on a Jacobian bound, but the paper does not clearly explain how this quantity is estimated in practice or how sensitive the method is to inaccuracies in this bound.
3. The experimental evaluation mainly considers two attack methods and does not examine adaptive attacks that explicitly target the proposed defense, leaving the robustness of LFPM under stronger threat models unclear.

---

> ### Author Rebuttal · Authors · 2026-03-31
>
> **Q1: Are The theoretical analysis  per-sample $x$ or in expectation over the data distribution?**
>
> **R1:** We thank the reviewer for this clarification.
>
> Our theoretical analysis (Theorem 3.1 and Corollary 1) is conducted in a parameter-space setting, where
> $f(\theta)$ denotes the model's feature mapping as a deterministic function of $\theta$, independent of any specific input $x$. **Thus, results are pointwise along $\theta(t)$ and do not explicitly involve $x$.**
>
> For standard learning, we consider empirical risk:
>
> $$
> \mathcal{L}(\theta) = \mathbb{E}_{x \sim \mathcal{D}}\left[\ell(x;\theta)\right]
> $$
>
> so gradients are:
>
> $$
> \nabla \mathcal{L}(\theta) = \mathbb{E}_{x \sim \mathcal{D}}\left[\nabla \ell(x;\theta)\right]
> $$
>
> Since expectation is linear, the gradient approximation result in Corollary 1 extends directly to the expected gradient under the data distribution. **For clarity, we omit the expectation in the derivation and focus on the deterministic parameter path behavior.**.
>
> **We will clarify the distinction between parameter-space and empirical-risk formulations in the revised manuscript.**
>
> ---
>
> **Q2: How is the Jacobian norm used to set the prompt perturbation radius in practice？**
> **R2:** We sincerely thank the reviewer for this insightful question. In Section 4.2, we map the feature deviation $\Delta z \in \{ v:\|v\|\le \rho \}$ to the prompt perturbation $\Delta P$, where the perturbation radius $r_{\rho}$ satisfies:
>
> $$
> r_{\rho} \le \frac{\rho}{\left\| J_P \right\|_2}.
> $$
>
> **Importantly**, the Jacobian norm is used only as a theoretical bridge between the prompt-space radius $r_{\rho}$ and feature bound $\rho$. **In practice, it is not explicitly computed**; we directly control $r_{\rho}$, which implicitly ensures the feature constraint $\rho$，making the method practical and grounded.
>
> **Sensitivity analysis:** We study $r_{\rho} \in [0.1,1.0]$, showing LFPM is stable for $r_{\rho} \in [0.1,0.5]$ with low ASR(T) (<1.0%). When $r_{\rho}=1.0$, ASR(T) rises to 10.55%, and to 24.64% at $r_{\rho}=10.0$, indicating overly large perturbations violate the feature constraint and degrade performance.
>
> --
>
> **Q3:  How robust is LFPM against potential adaptive attacks?**
>
> **R3:** We appreciate this important question. Recall that LFPM consists of two core mechanisms:
>
> - **Subspace Partitioning**, which disentangles adversarial perturbation and clean features;
> - **Anti-Backdoor Task Vector Generation**, interpreting SAM under CTL as robust optimization along the feature interpolation path.
>
> **Adaptive attack analysis:** An attacker aware of LFPM could adopt two strategies:
>
> - **Feature coupling attack:** enforce entanglement between clean and backdoor features (e.g., maximizing cosine similarity between backdoor and target-class clean features );
> - **Feature-interpolation-based backdoor loss (FI loss):** encourages the model to predict triggered inputs as the target class along the feature interpolation path (i.e., linear interpolation between clean and backdoor features). **Notably**, unlike similar mechanisms in BadMerging, we use clean pretrained features as reference, yielding a stronger and more general attack.
>
> **Importantly**, this objective is not specific to LFPM, but serves as a general enhancement for evaluating robustness under stronger adaptive attacks.
>
> **Results:** By combining BadMerging with the above strategies, results in Table 1 (supplementary PDF: https://anonymous.4open.science/r/paper_123-1540/PDF.pdf) show that, compared with the main manuscript, all baselines exhibit a significant increase in ASR(T) under adaptive attacks (e.g., SAU: 63.00%→99.20%, SAM: 10.03%→32.30%). In contrast, LFPM shows the smallest ASR increase (0.49%→2.38%), demonstrating strong resistance to adaptive attacks.
>
> **Conclusion:** This resilience is mainly attributed to LFPM's gradient accumulation mechanism, which  aggregates gradients from the backdoor-merged model to correct optimization directions, maintaining alignment with backdoor-relevant signals and enabling persistent backdoor removal under adaptive attacks.
>
> ---
>
> **Q4: How does LFPM’s orthogonality-based feature extraction extend to nonlinear classifier heads?**
>
> **R4:** We clarify that **the orthogonality result does not fundamentally rely on a linear prediction head, but follows from a more general output invariance condition**. For a general head $h$, a first-order expansion gives:
>
> $$
> h(g(x) + \Delta z) \approx h(g(x)) + J_h(g(x)) \Delta z.
> $$
>
> The output invariance condition then implies:
>
> $$
> J_h(g(x)) \Delta z \approx 0.
> $$
>
> i.e., $\Delta z$ lies in the null space of the Jacobian and does not affect prediction-relevant directions. **The linear case $h(z)=W^T z$ is a special instance, reducing to $\Delta z \perp \mathrm{span}(W)$, which simplifies the derivation.**
>
> **Thus, orthogonality is a special case of a general Jacobian-based invariance principle, which naturally extends to nonlinear heads and normalization-based architectures.**

---

> > ### Author Rebuttal · Reviewer_WhCz · 2026-04-01
> >
> > The author's feedback is clear, which solves my concerns.

---

> > > ### Author Response · Authors · 2026-04-02
> > >
> > > Thanks! It is our great pleasure to solve your concerns. Thanks for your positive comments.

---

### Official Review · Reviewer_gPYH · 2026-03-09

**Soundness:** 2
**Presentation:** 3
**Significance:** 2
**Originality:** 3
**Overall Recommendation:** 4
**Confidence:** 4

**Summary:**

This paper examines the security risks of model merging, especially vulnerability to backdoor attacks when combining task-specific models. The authors introduce Linear Feature Path Minimization (LFPM), a defense that mitigates backdoors during merging. LFPM formulates merging in feature space under a Cross-Task Linearity (CTL) assumption, using an anti-backdoor task vector to counter malicious components. The method involves gradient accumulation and loss path-integral strategies along interpolation paths. Evaluations show LFPM reduces ASR while maintaining clean task performance under both full and parameter-efficient fine-tuning settings.

**Compliance With Llm Reviewing Policy:**

Affirmed.

**Final Justification:**

Most of my concerns have been resolved by the authors.

**Key Questions For Authors:**

1. Can the authors provide empirical analysis validating CTL across different models and tasks?
2. How would LFPM perform if the adversary is aware of the defense and designs backdoors specifically to evade feature-path minimization?
3. Which adversaries are orthogonal and not orthogonal for this method? Can you provide a few experiments to clarify your statements on the orthogonality of adversaries affects the clean task performance from merged models ?
4. Have the authors tested LFPM on larger architectures (e.g., ViT, LLM adapters)?
5. Why is the area under the ASR curve (Path-ASR) a meaningful metric if only the final merged model is used at inference time?

**Limitations:**

The authors should discuss the conditions under which the proposed LFPM approach may fail.

**Strengths And Weaknesses:**

Strengths
- Overall, the structure of this paper is well organized and easy to follow.
- Unlike most works on merging in parameter space, this paper analyzes feature trajectories, which may better reflect representation learning behavior.
- The paper also demonstrates well its practicality on different configurations (i.e., PEFT, full finetuning) and other model merging methods (e.g., Simple Average, Ties-Merging)

Weaknesses
- The paper seems to assume that adversarial perturbations used to create the anti-backdoor tasks must be orthogonal to the clean semantic space. Otherwise, the performance on the clean task might suffer. In real-world scenarios, this assumption appears to be impractical since backdoor attacks tend to vary significantly, even with out-of-distribution samples [1].
- While the construction of defense mechanisms is well-established, the paper does not specify the creation and features of shadow datasets used to remove backdoor attacks from harmful adapters. The paper also lacks clear discussion on choosing and designing effective shadow datasets for migration from backdoor attacks, aside from experimental results.
- The choice of Path-ASR as a metric is not sufficiently justified. Since inference only uses the final merged model, it is unclear to me why the area under the ASR curve along the interpolation path is a meaningful indicator of backdoor robustness.
- No analysis on the extra computational cost from feature-space optimization and adversarial sample generation, especially when combining many task-specific models.

[1] Lyu, W., Yao, J., Gupta, S., Pang, L., Sun, T., Yi, L., ... & Chen, C. (2025). Backdooring Vision-Language Models with Out-Of-Distribution Data. In Proceedings of the 13th International Conference on Learning Representations.

---

> ### Author Rebuttal · Authors · 2026-03-31
>
> **Q1: Can the authors provide empirical analysis validating CTL across different models and tasks?**
>
> **R1:** We thank the reviewer for this question. **We kindly refer the reviewer to our response to Reviewer mWgG,  Q1**, where we provide additional empirical validation of CTL on a stronger architecture (ViT-L/14) and across different tasks. The results consistently confirm the validity of CTL.
>
> ---
>
> **Q2: How robust is LFPM against adaptive attacks?**
>
> **R2:** We thank the reviewer for this question. **We kindly refer the reviewer to our response to Reviewer WhCz, Q3**, where LFPM is evaluated under adaptive attacks combining feature-coupling and interpolation-aware strategies, showing strong robustness.
>
> ---
>
> **Q3: How does adversary orthogonality affect clean performance in merged models?**
>
> **R3:** We thank the reviewer for this insightful comment. We clarify the role of orthogonality in LFPM from four aspects:
>
> - **Purpose of orthogonality.** The orthogonality constraint is primarily introduced to preserve clean task performance during adversarial feature mining. As shown in Lemma 4.1 and Appendix E.7, removing it leads to a consistent drop in clean accuracy (avg. -3.50% CA drop). Moreover, prior work[1] suggests that backdoor and clean tasks tend to exhibit approximate orthogonality.
> - **Not a strict assumption.** We emphasize that orthogonality is not a required assumption for LFPM, but rather an idealized condition for theoretical analysis. In practice, low-orthogonality attacks may exist, where adversaries explicitly encourage strong entanglement between clean and backdoor features.
> - **Robustness under low-orthogonality attacks.** LFPM remains effective due to its second-stage SAM-like optimization along the feature interpolation path. In low-orthogonality scenarios, strong entanglement between backdoor and clean gradients makes direct separation ineffective; thus, we introduce a **Gradient Accumulation Mechanism**, which aggregates gradients from the merged model to approximate interpolation-path gradient directions, capturing backdoor-relevant signals and mitigating feature entanglement during the update of \$\\theta\_{K+1}\$   (**see Reviewer  WhCz, Q3 for adaptive robustness evidence**).
> - **Effect on clean performance.** We evaluate three settings: (i) with orthogonality, (ii) without it, and (iii) projecting perturbations onto clean subspace $span(W)$. Results consistently show that enforcing orthogonality improves clean accuracy (58.70% vs. 55.20% vs. 52.94%), while LFPM remains effective even when the constraint is relaxed.
>
> ---
>
> **Q4: Have the authors tested LFPM on larger architectures (e.g., ViT, LLM adapters)?**
>
> **R4:** Thanks. To evaluate scalability, we further test LFPM on a larger backbone, ViT-L/14, **with results in Table 2 (supplementary PDF: https://anonymous.4open.science/r/paper_123-1540/PDF.pdf)**. LFPM remains effective at a larger scale, achieving ASR(T) 0.89% and CA 61.93%, showing strong scalability.
>
> We also note that LFPM is compatible with PEFT paradigms, further suggesting its applicability to larger foundation models (e.g., LLMs);**please Reviewer 5Hoe Q2 for extensibility details**.
>
> ---
>
> **Q5: Why is Path-ASR meaningful if only the final merged model is used?**
>
> **R5:** We thank the reviewer for this valuable question. Path-ASR is introduced for two reasons:
>
> - **Practical consideration.** Existing works typically evaluate robustness at a single interpolation point (i.e., a fixed $\alpha$), while LFPM considers the full interpolation path ($\alpha \in [0,1]$). Since the optimal $\alpha$ is often unknown and varies across settings, single-point evaluation is insufficient.
> - **Comprehensive evaluation.** Path-ASR captures the cumulative backdoor effect along the interpolation path, providing a more holistic robustness measure. A lower Path-ASR indicates stronger and more consistent mitigation even when the model is closer to the backdoored source (i.e., small $\alpha$).
>
> **Overall**, Path-ASR serves as a stress-test metric under interpolation uncertainty in model merging.
>
> ---
>
> **Q6: How should shadow datasets be designed for effective backdoor removal?**
>
> **R6:** The shadow dataset acts as a proxy to broadly cover the model’s feature space. **An effective design should ensure** (i) class diversity, (ii) rich heterogeneous visual features, and (iii) general-purpose, non-domain-specific data. We follow these principles using public datasets (e.g., ImageNet1k).
>
> **We further validate on CIFAR-100 as a shadow dataset**, where LFPM achieves CA 68.48%, ASR(T) 0.48%, ASR(N) 6.15%, demonstrating robustness to dataset choice.
>
> ---
>
> **Q7: What is the computational cost of LFPM and scalability with task-specific models?**
>
> **R7:** **For a detailed comparison and cost analysis, we kindly refer the reviewer to Reviewer 5Hoe, Q1**.
>
> # References
>
> [1] K. Zhang *et al*., *“Exploring the orthogonality and linearity of backdoor attacks,”* IEEE S&P, 2024.

---

> > ### Author Rebuttal · Reviewer_gPYH · 2026-04-02
> >
> > I appreciate the authors' explanation, which addresses most of my concerns. In particular, they provide further discussion on how adversary orthogonality affects clean performance in merged models. The authors also include additional experimental results, testing LFPM on larger architectures and analyzing computational cost. In light of these improvements, I am willing to raise my score to a Weak Accept.

---

> > > ### Author Response · Authors · 2026-04-04
> > >
> > > We sincerely thank the reviewer for their positive and constructive feedback, as well as the time and effort spent on reviewing our work.

---

### Official Review · Reviewer_mWgG · 2026-03-10

**Soundness:** 2
**Presentation:** 3
**Significance:** 3
**Originality:** 2
**Overall Recommendation:** 4
**Confidence:** 3

**Summary:**

This paper investigates the backdoor attack problem in model merging scenarios. In this setting, multiple task-specific models are merged into a unified model without access to the original training data. Malicious participants can submit task models containing backdoors, causing trigger samples to be misclassified into a specified category within the merged model. To mitigate this issue, the authors propose the LFPM method. This approach shifts the task arithmetic from parameter space to feature space and leverages CTL properties for modeling. LFPM extracts adversarial features through subspace partitioning and optimizes anti-backdoor task vectors along feature interpolation paths. Experimental results across multiple datasets demonstrate that LFPM effectively reduces attack success rates while maintaining robust performance on legitimate tasks.

**Compliance With Llm Reviewing Policy:**

Affirmed.

**Key Questions For Authors:**

1. This method relies on the CTL property to support the optimization process in the feature space. Could the authors provide further empirical analysis to demonstrate the validity of this property across different model architectures or datasets?
2. How robust is this method against adaptive attacks when attackers can perceive the defense mechanism of LFPM? For instance, could attackers design specific backdoor trigger patterns to deliberately target the feature extraction stage and circumvent this defense mechanism?
3. The method introduces a multi-step optimization process and additional components. Could the authors provide a clearer efficiency analysis, such as comparisons with existing defense methods (e.g., IBVS or SAU) in terms of training time or memory overhead?

**Limitations:**

1. This method is theoretically based on the CTL assumption and the property of feature interpolation. For model architectures that deviate from this paradigm, its applicability and stability still need to be further verified.
2. Insufficient comparison of ablation experiments and related models in terms of temporal and spatial efficiency.

**Strengths And Weaknesses:**

Strengths
1. This paper addresses backdoor attacks in model fusion, a growing security concern in multi-task model deployment with significant practical relevance.
2. The proposed method mitigates backdoor attacks by focusing on the feature space, offering a novel and insightful alternative to existing approaches primarily based on the parameter space.
3. This paper introduces a two-stage defense that mitigates backdoor effects without sacrificing clean-task performance. Unlike prior approaches, it does not require input-space adversarial perturbations or explicit parameter-space interpolation during training, and it seamlessly integrates with the loss path-integral mechanism.
4. Experiments across multiple datasets, attack types, and fusion algorithms validate the effectiveness of the method to a certain extent.

Weaknesses
1. The paper's methodology relies heavily on the core CTL assumption that linear interpolation in parameter space corresponds to linear transformations in feature space. However, this assumption may not always hold in practical applications. For instance, this linear relationship could be disrupted in more complex network architectures, deeper models, or when merging heterogeneous model checkpoints from different sources. The paper currently lacks sufficient analysis of scenarios where this assumption may fail and does not discuss the method's stability and applicability in such cases.
2. From an overall conceptual perspective, the innovative differences between this work and existing task-vector-based defense methods appear relatively limited. Previous research has already modeled backdoors as task vectors in parameter space and mitigated them through vector operations. Although this paper reformulates relevant operations in the feature space, the fundamental conceptual differences from existing methods and practical advantages remain inadequately clarified, resulting in somewhat limited overall innovation.
3. Experimental evaluations primarily focus on standard attack settings like BadMerging and LoBAM, without exploring adaptive attack scenarios. If attackers can perceive LFPM's defense mechanism, they may design new trigger patterns or training strategies to evade the separation process in the feature space. The absence of such experiments leaves uncertainty regarding the method's robustness under stronger threat models.
4. At the implementation level, LFPM involves multiple stages, such as adversarial feature extraction, prompt optimization, and path integral-based optimization, which somewhat increases the overall process complexity. While the paper briefly mentions efficiency considerations, it lacks a systematic analysis of computational overhead and does not compare time or resource consumption with simpler defense methods.
5. Some key components of the method remain less intuitive in their explanation. For instance, modules like feature subspace partitioning and prompt space perturbation are presented in a highly technical manner, with limited discussion on their design motivations or intuitive understanding (possibly constrained by space?). Readers unfamiliar with prompt tuning or feature space optimization may find these sections challenging to grasp.
6. Furthermore, current experiments are primarily conducted on the CLIP architecture and its visual encoder. Consequently, the method's applicability to other model structures or cross-modal tasks remains unclear. The absence of cross-architecture or cross-modal validation leaves the method's generalization capabilities to be further confirmed.

---

> ### Author Rebuttal · Authors · 2026-03-31
>
> **Q1: Does CTL property generalize across different model architectures and datasets?**
>
> **R1:** We thank the reviewer for the thoughtful question. We clarify that CTL is defined under a standard setting where models share the same architecture and are fine-tuned from a common pretrained initialization.
>
> **The original CTL study [1]** validates this property across diverse pretrained models, including vision models (ViT-B/32, ViT-L/14) on 8 datasets (Cars, DTD, EuroSAT, etc) and language models (T5) on 6 NLP benchmarks (IMDB, RACE, QASC, etc.).
>
> **In our work**, LFPM focuses on vision tasks; we verify the CTL on ViT-B/32 in Appendix E.4, and **further conduct additional evaluations under two settings:**
>
> - stronger architectures such as ViT-L/14, and
> - CIFAR-100 shadow dataset.
>
> Results (**Fig. 1 and 2 in supplementary PDF: https://anonymous.4open.science/r/paper_123-1540/PDF.pdf**) show CTL remains valid: under setting (1),  it holds in a restricted range (e.g., $\alpha \in [0.6, 1.0]$), while Under setting (2), it holds over a broader range (e.g., $\alpha \in [0.5, 1.0]$), covering our setting ($\alpha_2=0.9$).
>
> **Overall**, these results confirm CTL’s validity under both settings, supporting LFPM’s applicability within the CTL framework.
>
> ---
>
> **Q2: How robust is LFPM against adaptive attacks?**
>
> **R2:**  **We kindly refer the reviewer to our response to Reviewer WhCz,  Q3**, where LFPM is evaluated under adaptive attacks combining feature-coupling and interpolation-aware strategies. The results show that LFPM remains robust.
>
> ---
>
> **Q3: A systematic analysis of computational cost for defense methods**
>
> **R3:**  Thanks.  **We refer the reviewer to our response to Reviewer 5Hoe, Q1 for a detailed comparison.**
>
> ---
>
> **Q4: How does LFPM differ from prior task-vector defenses, and what are the benefits of its unified feature-space formulation？**
>
> **R4:** Thanks. LFPM differs from existing task-vector-based defenses in three aspects:
>
> - **Clean–backdoor trade-off.** Existing methods struggle to balance clean accuracy and backdoor removal due to entangled task and backdoor features. In contrast, LFPM uses subspace partitioning to disentangle them, enabling effective removal while preserving clean performance.
> - **Interpolation-path robustness.** Prior work typically focuses on a single merged model (i.e., fixed coefficient $\alpha$). LFPM instead enforces robustness over the full interpolation path ($\alpha \in [0,1]$) via SAM-like optimization along the feature interpolation path.
> - **Theoretical grounding.** Existing methods are largely empirical, motivated by observed task-vector similarity across attacks. In contrast, LFPM is grounded in CTL and SAM theory, providing a principled, robust optimization view.
>
> Unified feature-space reformulation offers two advantages:
>
> - **Algorithmic simplification.** It removes the need for input-space perturbations and explicit parameter interpolation during optimization.
> - **A unified perspective.** It provides a shared feature-space view linking attack and defense, and can be combined with adversarial feature augmentation [2] to further improve robustness.
>
> ---
>
> **Q5:How intuitive and well-motivated are subspace partitioning and prompt-space perturbation?**
>
> **R5**: Thanks.
>
> **Design motivation.** Existing methods introduce  input-space perturbations (e.g., SAU) or rely on trigger inversion (e.g., PAM), which may distort semantics. In contrast, LFPM uses learnable prompts to encode perturbations, preserving semantic structure while enabling adversarial modeling. Subspace partitioning separates clean semantic features from backdoor components, reducing interference and improving disentanglement for clean-task preservation.
>
> **Intuition and advantage.** Prompts act as lightweight carriers of adversarial signals independent of inputs, while subspace partitioning decomposes representations into clean/adversarial components. This makes Anti-backdoor Task Vector optimization more tractable: within CTL, feature deviations are modeled in a low-dimensional prompt space rather than full parameter space, leading to a more stable and interpretable optimization process.
>
> **We will further clarify these intuitions and motivations in the revised version**.
>
> ---
>
> **Q6: How generalizable is LFPM beyond CLIP, including across architectures and cross-modal settings?**
>
> **R6:** Thanks. LFPM is a general feature-space optimization method and is not tied to CLIP (**please see Reviewer 5Hoe Q2**). We validate it on ViT-L/14 with consistent gains (**see Reviewer gPYH Q4**), and leave broader extensions as future work.
>
> ---
>
> # References
>
> [1]Z. Zhou et al., “On the emergence of cross-task linearity in the pretraining-finetuning paradigm,” ​arXiv:2402.03660​, 2024.
>
> [2]T. Chen et al., *“Adversarial feature augmentation and normalization for visual recognition,” arXiv:2103.12171, 2021.

---

> > ### Author Rebuttal · Reviewer_mWgG · 2026-04-04
> >
> > I appreciate the authors' detailed response.

---

> > > ### Author Response · Authors · 2026-04-04
> > >
> > > We thank the reviewer for the positive feedback and for acknowledging that the concerns have been adequately addressed.

---

### Decision · Program_Chairs · 2026-04-30

**Decision:**

Accept (regular)

**Comment:**

This paper proposes LFPM, a backdoor defense framework for model merging that shifts task arithmetic from parameter space to feature space under the Cross-Task Linearity (CTL) framework, introducing an anti-backdoor task vector optimized via gradient accumulation and a loss path-integral mechanism.

On the one hand, reviewers broadly recognized the paper's novel feature-space perspective, solid theoretical grounding, and practical relevance to model merging security. After the rebuttal, Reviewers mWgG and gPYH fully resolved their concerns and maintained or raised their scores, and Reviewer WhCz was also fully satisfied. The authors provided additional empirical validation of CTL, adaptive attack evaluations, efficiency analysis, and supplementary CNN experiments, which addressed the majority of concerns raised.

On the other hand, Reviewer 5Hoe remained only partially resolved, noting that the cross-architecture evaluation provided (a single ResNet-50 experiment in a non-merging setting) is limited and falls short of demonstrating the generality of LFPM in realistic model merging scenarios beyond CLIP-style models. The evaluation scope of the main paper remains primarily restricted to CLIP-ViT-B/32, and the broader applicability to other architectures and modalities has not been substantiated with sufficient empirical evidence in the main manuscript.

Balancing these considerations, I lean toward accepting this paper given the largely positive post-rebuttal assessments, while acknowledging that the limited cross-architecture validation remains a weakness that constrains the paper's overall significance.